# Schizophrenia-related Xpo7 haploinsufficiency leads to behavioral and nuclear transport pathologies

Saori Toyoda[1,4], Masataka Kikuchi[2,4], Yoshifumi Abe [ID][3], Kyosei Tashiro[3], Takehisa Handa[1], Shingo Katayama [ID][1], Yukiko Motokawa[1], Kenji F Tanaka[3], Hidehiko Takahashi[1] & Hiroki Shiwaku [ID][1][✉]

## Abstract

**Recent genetic studies by the Schizophrenia Exome Sequencing Meta-Analysis (SCHEMA) consortium have identified that protein-truncating variants of exportin 7 (XPO7) can increase the risk of schizophrenia (odds ratio, 28.1). Here we show that mice with Xpo7 haploinsufficiency (Xpo7$^{+/-}$ mice) present with cognitive and social behavioral impairments. Through proteome analysis using immunoprecipitation and frontal cortex nuclear isolation of Xpo7$^{+/-}$ mice, we identify 45 molecules interacting with Xpo7, including CutC, Rbfox3, and Gria3. Through single-nucleus RNA sequencing of the frontal cortex and striatum of Xpo7$^{+/-}$ mice differentiating between the onset and progressive stages, we also identify 284 gene expression changes that correlate with these stages. These genes encompass high-odds risk genes of schizophrenia identified by SCHEMA, including Gria3, Grin2A, Herc1, and Trio. Furthermore, our approach reveals 15 gene expression changes in the frontal cortex that correlate with the progressive stages. Our findings indicate the importance of investigating whether the interactions among the high-risk genes identified by SCHEMA contribute to a common schizophrenia pathology and underscore the significance of stage-dependent analysis.**

**Keywords** Schizophrenia; Xpo7; SCHEMA; Nuclear Transport; Progressive Pathologies
**Subject Categories** Molecular Biology of Disease; Neuroscience

## Introduction

Schizophrenia is a psychiatric disorder characterized by hallucinations, delusions, cognitive decline, and social impairment. It is caused by diverse genetic backgrounds. Researchers have generated and analyzed genetically engineered mice with schizophrenia-related variants using data from genetic studies in affected families and high-risk genomic regions. Analyses of exomes from over 20,000 individuals diagnosed with schizophrenia revealed ten genes

associated with significantly increased odds ratios for protein-truncating variants (PTVs), indicative of probable loss of function (Singh et al, 2022). These genes include *SETD1A*, *XPO7*, *CUL1*, *GRIA3*, *SP4*, *GRIN2A*, *HERC1*, *TRIO*, *RB1CC1*, and *CACNA1G* (Singh et al, 2022). While haploinsufficiency mice models for Setd1a and Grin2a have been explored within schizophrenia pathophysiology (Chen et al, 2022; Farsi et al, 2023; Mukai et al, 2019; Nagahama et al, 2020), further investigation into these genes' cellular and molecular pathology is crucial for understanding the pathology of schizophrenia in light of the disease's heterogeneous nature and shared pathologies among affected individuals. Mutations in high-risk genes may potentially trigger shared pathologies observed in other patients with schizophrenia.

In addition to investigating the molecular pathogenesis of high-risk gene variants in schizophrenia, understanding the molecular mechanisms underlying the progressive pathogenesis of schizophrenia is also important. Studies report that if the duration of the untreated period (DUP) is long, patients become refractory to treatment, suggesting the existence of progressive pathogenesis in schizophrenia (Marshall et al, 2005; Perkins et al, 2005). It has also been reported that there is a subtype of patients showing progressive cognitive decline over 10 years in longitudinal studies (Fett et al, 2020; Starzer et al, 2024; Zanelli et al, 2019). Those with poor or declining cognition were less likely to participate in follow-up assessments, suggesting that some patients with worse cognitive decline during the course exist (Starzer et al, 2024). Relapse in schizophrenia is also associated with disease progression in terms of treatment resistance and a general decrease in brain volume (Andreasen et al, 2013; Emsley et al, 2013). Thus, we hypothesize the existence of a progressive molecular pathogenesis. Among high-risk gene products, hypothesis of progressive pathogenesis can be particularly important for XPO7, a gene product in nuclear transport regulation (Aksu et al, 2018). Neurons may be especially vulnerable to disruptions in nuclear transport due to their non-dividing nature and the absence of typical nuclear membrane loss observed in mitotic cells.

Herein, we present our analysis of Xpo7 haploinsufficiency mice, focusing on the molecular progressive pathology comparing the onset (3 months of age) and chronic phases (6 months of age). We conducted a study to determine if Xpo7 haploinsufficiency mice show cognitive and social behavioral impairments. In

[1]Department of Psychiatry and Behavioral Sciences, Institute of Science Tokyo, 1-5-45, Yushima, Bunkyo-ku, Tokyo 113-8510, Japan. [2]Department of Molecular Genetics, Brain Research Institute, Niigata University, 1-757 Asahimachidori, Chuo-ku, Niigata 951-8585, Japan. [3]Division of Brain Sciences, Institute for Advanced Medical Research, Keio University School of Medicine, Tokyo, Japan. [4]These authors contributed equally: Saori Toyoda, Masataka Kikuchi. [✉]E-mail: shiwaku.npat@mri.tmd.ac.jp

addition, we carried out a proteomic analysis to identify molecular changes in nuclear transport caused by Xpo7 haploinsufficiency. We performed single-nucleus RNA sequencing analysis to uncover molecular and cellular signatures associated with the progressive features of Xpo7 haploinsufficiency. These analyses revealed the relationship between Xpo7 haploinsufficiency and other high-risk gene products.

# Results

## Xpo7 haploinsufficiency mice show cognitive and social behavioral impairments

We employed the CRISPR-Cas9 technique to delete the 7th exon of Xpo7 (Fig. 1A). The removal of exon 7 was confirmed through genome sequencing and genotyping Polymerase Chain Reaction (Fig. EV1A). Xpo7 knockout mice ($Xpo7^{-/-}$) have a low birthrate, and most do not survive past one month. This partially penetrant embryonic lethality aligns with a previous study (Modepalli et al, 2022). Analysis using western blot and liquid chromatography–mass spectrometry (LC-MS/MS) on 1-month-old mice showed the absence of Xpo7 expression in $Xpo7^{-/-}$ mice, with a significant reduction (approximately halved) in Xpo7 expression levels in heterozygous knockout ($Xpo7^{+/-}$) mice compared to wild type ($Xpo7^{+/+}$) littermates (Fig. 1A,B). Xpo7 is a 1088 amino acid (aa) molecule, and exons 1–6 correspond to 1–199 aa, exon 7 to 200–255 aa, and exons 8–28 to 256–1088 aa. Because the antigen for the anti-Xpo7 antibody used in the western blot in Fig. 1B recognizes 520–594 aa, exons 7–28 are not expressed as proteins. In LC-MS/MS, six types of peptide fragments were detected from the sequence including 1–199 aa (exons 1–6) from $Xpo7^{+/+}$ and $Xpo7^{+/-}$ mice, but not from $Xpo7^{-/-}$ mice (Fig. EV1B). Collectively, these data suggest that the truncated proteins of exons 1–6 are not expressed in the $Xpo7^{+/-}$ and $Xpo7^{-/-}$ mice analyzed in this study.

There was no obvious change in brain size in hematoxylin and eosin (H&E) staining, and no apparent disturbance of the cortical layer structure by immunohistochemical analysis using layer markers between $Xpo7^{+/+}$ and $Xpo7^{+/-}$ mice (Fig. EV1A,B). We also performed Golgi staining to analyze the dendrites and spines. A significant decrease in the apical and basal dendrites of L2/3 neurons and the apical dendrites of L5/6 neurons in the frontal cortex was observed (Fig. EV2A,B). There were no significant differences in spine density per dendrite length (Fig. EV3A,B).

We conducted a study to determine if $Xpo7^{+/-}$ mice display behavioral pathologies. The results showed that $Xpo7^{+/-}$ mice exhibited cognitive deficits in the Y maze test at 3 and 6 months (Fig. 1D). The onset of cognitive deficits was defined at 3 months of age, as $Xpo7^{+/-}$ mice showed normal cognitive function at 2 months (Fig. EV4A). In addition, $Xpo7^{+/-}$ mice showed deficits in pre-pulse inhibition at both 3 and 6 months of age, which is a recognized endophenotype of schizophrenia (Fig. 1E) (Powell and Miyakawa, 2006; Powell et al, 2009; Swerdlow et al, 2008). In the three-chamber test, impaired social novelty preference was significantly observed in $Xpo7^{+/-}$ mice at 6 months of age, whereas it was not obvious at 3 months of age, indicating a possible progressive change in phenotype (Fig. 1F). There was a slight tendency toward impaired social novelty preference at 3 months of age. To verify that the impaired social novelty preference was independent of

memory deficits, we conducted a novelty object recognition test, which showed no difference between $Xpo7^{+/-}$ mice and $Xpo7^{+/+}$ mice (Fig. 1G). $Xpo7^{+/-}$ mice exhibited normal locomotor activity and anxiety in the open field and elevated plus maze tests (Fig. EV4B,C).

## Identification of 45 molecules that interact with Xpo7 and are affected by Xpo7 haploinsufficiency in the frontal cortex of mice

We used the following method to identify molecules with abnormal nuclear import or export in the frontal cortex of $Xpo7^{+/-}$ mice. Based on the idea that molecules transported to or from the nucleus via Xpo7 bind to Xpo7, we performed immunoprecipitation analysis using $Xpo7^{+/+}$ and $Xpo7^{-/-}$ mice. This was followed by liquid chromatography–mass spectrometry (LC-MS/MS) analysis to isolate Xpo7-interacting molecules (Fig. 2A). At the same time, we isolated nuclei from the frontal cortex of $Xpo7^{+/-}$ and $Xpo7^{+/+}$ mice. We used LC-MS/MS analysis to quantify nuclear protein levels to identify molecules with significant changes in nuclear abundance (Fig. 2A). Through these methods, we could identify molecules that bind to Xpo7 and exhibit significant changes in nucleoprotein levels due to Xpo7 reduction in $Xpo7^{+/-}$ mice (Fig. 2A).

More detailed results are below. Initially, we searched for molecules that were most affected in the nuclear fraction of the frontal cortex of $Xpo7^{-/-}$ mice by LC-MS/MS analysis. Among the proteins detected in the nuclear fraction of the frontal cortex of $Xpo7^{-/-}$ mice, CutC was the most notably decreased (Fig. 2B). LC-MS/MS analysis also revealed a significant reduction (approximately half) in $Xpo7^{+/-}$ mice compared with that in $Xpo7^{+/+}$ littermates (Fig. 2C). Western blot analysis also confirmed the marked reduction of the CutC protein levels in the frontal cortex cell nuclei in $Xpo7^{-/-}$ mice and $Xpo7^{+/-}$ mice (Fig. 2D). These results indicate that CutC is transported into the nucleus by Xpo7 and is highly affected by Xpo7 haploinsufficiency.

To identify molecules that bind to Xpo7, we calculated a ratio by dividing the amount of each molecule that was immunoprecipitated by the anti-Xpo7 antibody from the frontal cortex of $Xpo7^{+/+}$ mouse by that of $Xpo7^{-/-}$ mice. Because the ratio for CutC was 1.7, we considered molecules that interacted with Xpo7 as those exhibiting a ratio equal to or exceeding 1.7. Consequently, we discovered 411 molecules that bind to Xpo7 (Fig. 2E). Of these, 45 showed significant ($P < 0.05$) changes in protein levels by LC-MS/MS analysis of the frontal cortex cell nuclei of $Xpo7^{+/-}$ mice compared with the $Xpo7^{+/+}$ mice (Fig. 2E).

The 45 molecules are presented in Fig. 3F. Gene Ontology (GO) analysis of these 45 molecules revealed that they contain many receptors and transporters involved in synapses (Fig. 3G). Using the STRING database, a 1-hop protein–protein interaction analysis with Xpo7 revealed 847 molecules (Fig. EV5A). Infomap algorithm and GO analysis showed that these 1-hop Xpo7 interacting molecules are significantly associated with transmembrane transport, immune response, axon guidance, synaptic signaling, and proteasome (Fig. EV5A,B).

Gria3, a constitutive subunit of the α-amino-3-hydroxy-5-methylisoxazole-4-propionic acid (AMPA) receptor, is one of these molecules. PTVs of GRIA3 have been identified as high-risk variants for schizophrenia from the exome analysis, as has XPO7

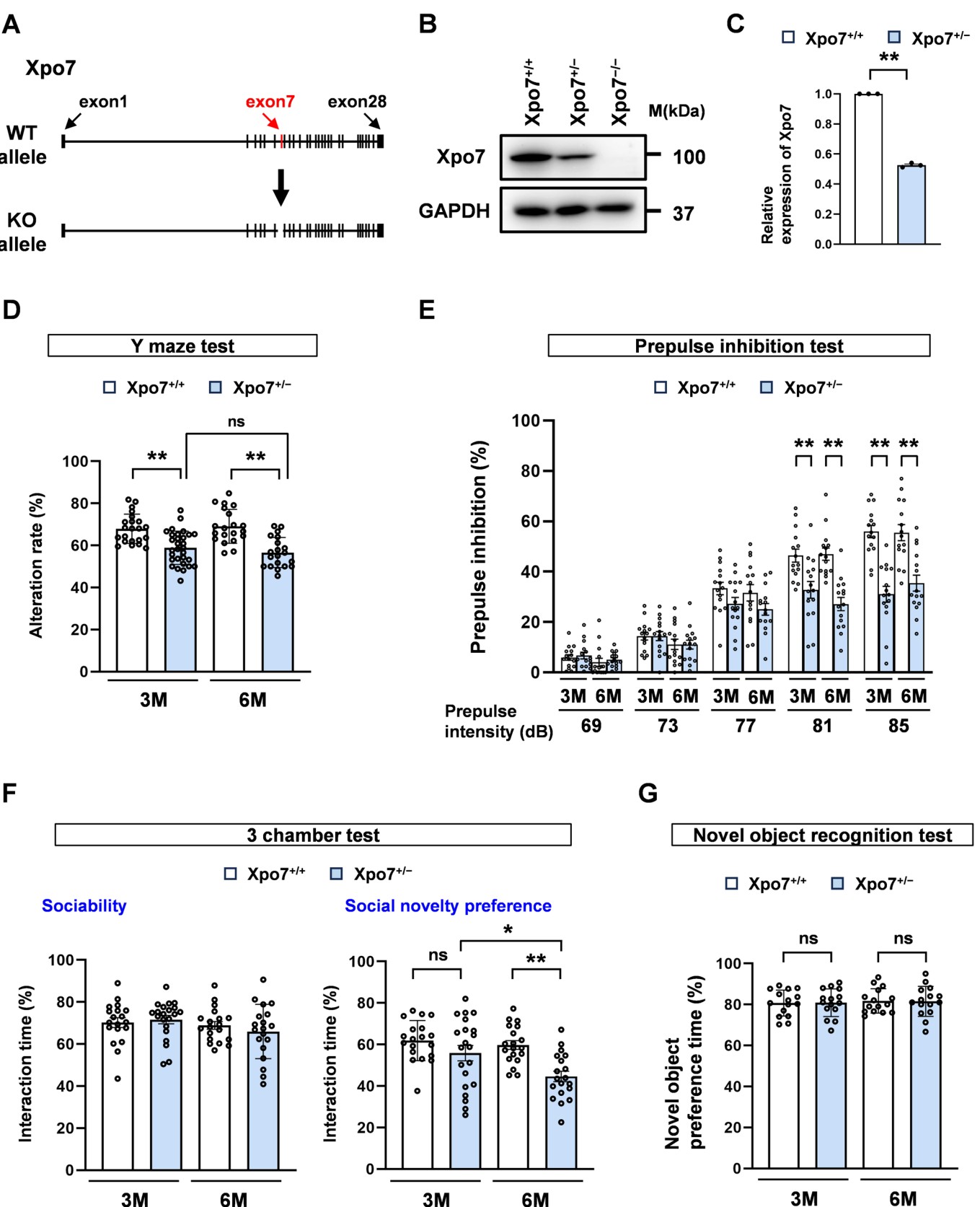

◄ **Figure 1. Xpo7$^{+/-}$ mice manifest cognitive and social behavioral impairments.**

(A) Schematic of Xpo7 knockout allele generation. (B) Western blot analysis of Xpo7 in the frontal cortex of Xpo7$^{+/+}$, Xpo7$^{+/-}$, and Xpo7$^{-/-}$ mice. (C) The relative expression of Xpo7 by proteome analysis of the frontal cortex in Xpo7$^{+/+}$ and Xpo7$^{+/-}$ mice. **$P < 0.01$ ($P = 0.00000539$, $N = 3$ experiments, three mice per group (1, 3, 6 months of age)/experiment; Tukey's HSD test). Specifically, three Xpo7$^{+/+}$ mice and three Xpo7$^{+/-}$ mice at 1 month of age were combined as one sample each to obtain the relative change of Xpo7$^{+/-}$ mice to Xpo7$^{+/+}$ mice. Similarly, three Xpo7$^{+/+}$ mice and three Xpo7$^{+/-}$ mice at 3 and 6 months of age were also analyzed to determine the relative change of Xpo7$^{+/-}$ mice to Xpo7$^{+/+}$ mice. Consequently, the relative change of Xpo7$^{+/-}$ mice to Xpo7$^{+/+}$ mice at 1, 3, and 6 months were statistically analyzed to identify the molecules that change consistently in Xpo7$^{+/-}$ mice regardless of their age in months. Data are expressed as the mean ± s.e.m. (D) Alteration ratios in the Y maze test with Xpo7$^{+/+}$ and Xpo7$^{+/-}$ mice at 3 and 6 months of age. **$P < 0.01$ (3 M $P = 0.000084$; 6 M $P = 0.000074$; 3 M Xpo7$^{+/-}$ vs 6 M Xpo7$^{+/-}$ $p = 0.31$; $N = 20$–30 mice per group, Tukey's HSD test). Data are expressed as the mean ± s.e.m. (E) Pre-pulse inhibition rates of Xpo7$^{+/+}$ and Xpo7$^{+/-}$ mice at 3 and 6 months of age. **$P < 0.01$ (81 dB 3 M $P = 0.0032$; 81 dB 6 M $P = 0.000006$; 85 dB 3 M $P = 0.000001$; 85 dB 6 M $P = 0.00013$; $N = 15$ mice per group, Tukey's HSD test). Data are expressed as the mean ± s.e.m. (F) Sociability and social novelty preference in the three-chamber test of Xpo7$^{+/+}$ and Xpo7$^{+/-}$ mice at 3 and 6 months of age. *$P < 0.05$, **$P < 0.01$ (3 M $P = 0.182$; 6 M $P = 0.000054$; 3 M Xpo7$^{+/-}$ vs 6 M Xpo7$^{+/-}$ $P = 0.0189$; $N = 10$–14 mice per group; Tukey's HSD test). Data are expressed as the mean ± s.e.m. (G) Novel object recognition test. There were no significant differences between the groups ($N = 15$ mice per group; Tukey's HSD test). Data are expressed as the mean ± s.e.m. Source data are available online for this figure.

(Singh et al, 2022). We observed elevated levels of Gria3 within the nuclei of frontal cortical cells in Xpo7$^{+/-}$ mice through LC-MS/MS analysis (Fig. 3A). Because the total Gria3 amount is compatible between Xpo7$^{+/-}$ mice to Xpo7$^{+/+}$ mice, we hypothesized that the extranuclear Gria3 levels are reduced, including the membrane fraction where Gria3 functions. We quantified the extranuclear Gria3 levels by western blotting analysis and confirmed a decrease in Gria3 outside the nucleus at 1, 3, and 6 months of age (Figs. 3B and EV6A).

Among the 45 molecules, others exhibited potential pathogenicity. Rbfox3, also known as NeuN, showed a 50% reduction within the nucleus under Xpo7 haploinsufficiency, similar to the effect observed in CutC (Fig. 3C,D). In addition, Slitrk5 and Fyn in the nucleus are upregulated in Xpo7$^{+/-}$ mice (Fig. 3E–G). Slitrk5 (Odds ratio 6.69, $P = 0.000917$) and Fyn (Odds ratio 10.7, $P = 0.000596$) were molecules linked to an increased susceptibility to schizophrenia (Singh et al, 2022). Western blotting analysis confirmed the reduction of nuclear Rbfox3 levels and extranuclear Slitrk5 and Igsf21 levels in Xpo7$^{+/-}$ mice at 3 months of age (Figs. 3D,F and EV6B). Alterations in Igsf21 expression could disrupt the balance between excitatory and inhibitory neurotransmission (Tanabe et al, 2017) (Fig. 3H). The group of 45 molecules also includes various constituent subunits of K-channels, Ca-channels, and GABA receptors (Figs. 3I and EV6C–F).

## Single-nucleus RNA sequencing analysis of frontal cortex excitatory neurons revealed the Xpo7 haploinsufficiency phenotype-associated gene expressions

The frontal cortex and the striatum are the brain areas that are significantly associated with schizophrenia (Chand et al, 2020; McCutcheon et al, 2019; Simpson et al, 2010). Cognitive function and social interactions are closely linked to the frontal cortex functionality. The dopamine pathology of schizophrenia is well known and the association between the striatum, which strongly expresses dopamine receptors, and schizophrenia has been repeatedly reported (Chand et al, 2020; McCutcheon et al, 2019; Simpson et al, 2010). Therefore, to determine the specific cell populations that are affected in the frontal cortex and striatum of Xpo7$^{+/-}$ mice, we conducted single-nucleus RNA sequencing analysis at 3 and 6 months in the sections containing both brain regions (Bregma 1.30–3.30, excluding the olfactory area) (Fig. 4A,B; Table EV1). The neurons in these sections can be dissected into

subpopulations in silico using single-nucleus RNA sequencing data. For example, the vast majority of excitatory neurons from these sections originate in the frontal cortex and these neurons can be further identified by cortical layer markers (Fig. 4D,E). In addition, inhibitory neurons in these sections originate in the frontal cortex and striatum and can be further classified by the dopamine D1 receptor (Drd1) and dopamine D2 receptors (Drd2) for medium spiny neurons (dSPN and iSPN) in the striatum as well as other markers, such as parvalbumin (Fig. 6A,B).

We detected differentially expressed genes (DEGs) in excitatory and inhibitory neurons at 3 and 6 months (Fig. 4C). To perform the subgroup analysis of excitatory neurons, we analyzed those extracted from layer 2/3 (L2/3: Stard8+ Calb1 +), layer 4 (L4: Rorb +), layer 5 (L5a: Deptor +, L5b: Tcerg1l+ Pou3f1 +, L5c: Nnat+ Abi3bp+ Htr2c +, L5d: Nxph1+ Tshz2), layer 5/6 (L5/6: Rgs12 +), and layer 6 (L6: Syt6 +) (Fig. 4D,E). There was no obvious difference in the ratio of neurons in the cortical layers between Xpo7$^{+/+}$ mice and Xpo7$^{+/-}$ mice (Fig. EV7A). In Xpo7$^{+/-}$ mice, multiple layers showed differentially expressed genes (DEGs) at 3 months of age, with L5c cells displaying more gene alterations (Fig. 4F). At 6 months, DEGs in L5c neurons exceeded that of other layers in Xpo7$^{+/-}$ mice (Fig. 4F).

We initially focused our analysis on L5c excitatory neurons. The GO analysis revealed distinct GO terms at 6 months in Xpo7$^{+/-}$ mice (Fig. EV7B). We then used the following approach to identify the DEGs related to the pathology of Xpo7$^{+/-}$ mice. It is important to note that among the DEGs in these excitatory neurons, we need to determine that the DEGs in Xpo7$^{+/-}$ mice implicated at the onset of pathology at 3 months not only differ from age-matched controls, but also the controls at 6 months. Similarly, DEGs in Xpo7$^{+/-}$ mice at 6 months of age should differ not only from age-matched controls, but also from controls at 3 months of age. Genes that meet these criteria are more closely related to the pathogenesis of Xpo7 haploinsufficiency. The analysis revealed 107 DEGs in L5c neurons (Fig. 4G). Among these genes, Mt-ATP6, Gpm6a, Ubash3b, and Nfib were identified as progression-associated genes (Fig. 4H).

The following DEGs in other cortical layers were also identified using this approach: 48 in L2/3 neurons (Fig. 5A), 52 in L4 neurons (Fig. 5B), 36 in L5a neurons (Fig. 5C), and 23 in L6 neurons (Fig. 5D). Among these molecules, Slc9a9 and Gabra2 were identified in L2/3 neurons, Slc9a9, Trps1, Prr16, Prkca, and Smoc2 in L4 neurons, Mef2c, Homer1, Phka2, and Guf1 in L5a neurons,

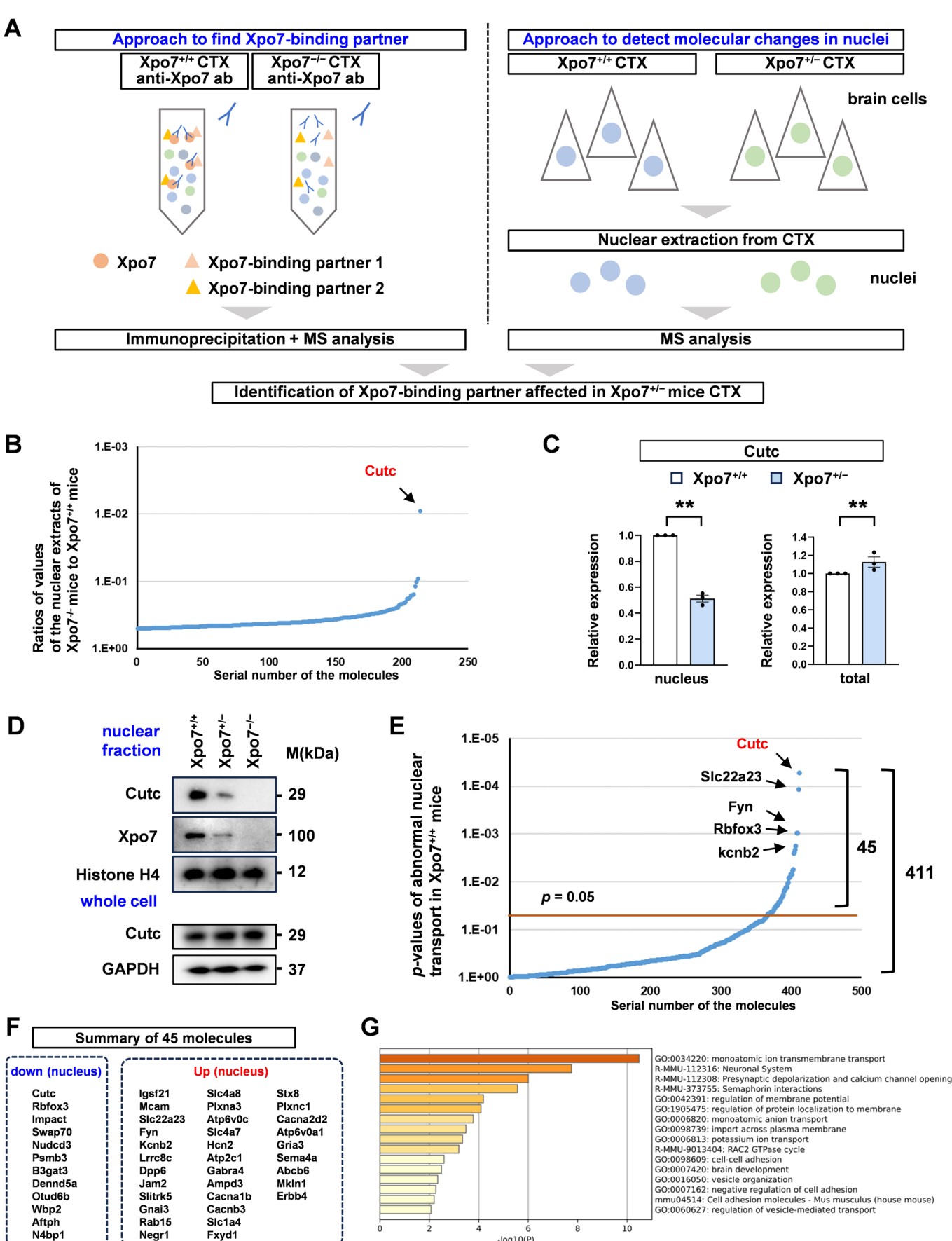

**Figure 2. Identification of molecules with affected nuclear transport in Xpo7$^{+/-}$ mice.**

(A) Experimental protocol to identify Xpo7-binding partners affected in the frontal cortex of Xpo7$^{+/-}$ mice. (B) Ratios of values of the nuclear extracts of Xpo7$^{-/-}$ mice to Xpo7$^{+/+}$ mice at 1 month of age. The nuclear extracts were obtained from three Xpo7$^{-/-}$ mice and three Xpo7$^{+/+}$ mice and analyzed by LC-MS/MS. (C) LC-MS/MS analysis showing the relative expression of CutC in nuclei and whole cells from the frontal cortex of Xpo7$^{+/+}$ and Xpo7$^{+/-}$ mice. **$P < 0.01$ (nucleus $P = 0.000052$; total $P = 0.087$; $N = 3$ experiments, 3 mice per group/experiment; Tukey's HSD test). As described in Fig. 1C, three Xpo7$^{+/+}$ mice and three Xpo7$^{+/-}$ mice at 1 month of age were combined as one sample each to obtain the relative change of Xpo7$^{+/-}$ mice to Xpo7$^{+/+}$ mice. Similarly, three Xpo7$^{+/+}$ mice and three Xpo7$^{+/-}$ mice at 3 and 6 months of age were also analyzed to determine the relative change of Xpo7$^{+/-}$ mice to Xpo7$^{+/+}$ mice. Consequently, the relative change of Xpo7$^{+/-}$ mice to Xpo7$^{+/+}$ mice at 1, 3, and 6 months were statistically analyzed to identify the molecules that change consistently in Xpo7$^{+/-}$ mice regardless of their age in months. Data are expressed as the mean ± s.e.m. (D) Western blot analysis of CutC in nuclei and whole cells from the frontal cortex of Xpo7$^{+/+}$, Xpo7$^{+/-}$, and Xpo7$^{-/-}$ mice at 1 month of age. (E) $P$ values of Xpo7-binding molecules for abnormal nuclear transport by proteome analysis in Xpo7$^{+/-}$ mice. Among Xpo7-binding molecules, 411 molecules detected by proteome analysis are plotted. ($N = 3$ experiments, three mice per group/experiment; Tukey's HSD test). (F) Summary of 45 molecules identified in (E). (G) GO analysis of the 45 molecules identified in (E). Source data are available online for this figure.

and Runx1t1 in L6 neurons were found to be associated with disease progression (Fig. 5E).

## Single-nucleus RNA sequencing analysis of striatum inhibitory neurons reveals the Xpo7 haploinsufficiency phenotype-associated gene expressions

We then performed a subcluster analysis of inhibitory neurons, including cortical and striatal inhibitory neurons. The latter were broadly classified into direct spiny projection neurons (dSPN), indirect spiny projection neurons (iSPN), and eccentric spiny projection neurons (eSPN) (Fig. 6A,B). Our analysis uncovered significant DEGs in iSPN and dSPN at 3 and 6 months in Xpo7$^{+/-}$ mice (Fig. 6C,D).

As shown in Figs. 4H and 5A–D, we found 57 DEGs in dSPN and 62 in iSPN (Fig. 6E,F). Abi3bp and Cntn4 in dSPN and Wipf3 and Npas3 in iSPN were identified as progression-associated molecules among these genes. Furthermore, 26 genes overlap between dSPN and iSPN DEGs, including Grin2A and Herc1. Trio was included in the DEGs specific to dSPN (Figs. 6G and EV7C). High-risk variants for schizophrenia were identified in the PTVs of Grin2A, Herc1, and Trio through exome analysis (Singh et al, 2022).

## DEGs identified through single-nucleus RNA sequencing demonstrate significant overlap with SCHEMA molecules

We found 215 disorder-associated DEGs in cortical layers, plus 93 in the striatum (Figs. 4G and 5A–D for cortical layers; Figs. 6E–G for the striatum). After removing duplicates, this yielded a total of 284 unique genes. We examined the overlap between the total count of disorder-associated DEGs in cortical layers and striatum and ten genes identified from the exome analysis (referred to as SCHEMA molecules) (Singh et al, 2022). We found four genes (Gria3, Grin2A, Herc1, Trio) that overlapped between these clusters, with an increase of 21.1-fold compared to the expected value (Fig. 7).

## Discussion

In this study, we demonstrated that mice with Xpo7 haploinsufficiency present with cognitive and social behavioral impairments. In addition, we identified 45 molecules that interact with Xpo7 and are affected by Xpo7 haploinsufficiency (referred to as Xpo7 direct-

effect molecules). Analysis using single-nucleus RNA sequencing revealed DEGs associated with the onset and progression in the frontal cortex excitatory neurons and striatal dSPN and iSPN in Xpo7$^{+/-}$ mice. Furthermore, there were significant overlaps between SCHEMA molecules and disease-associated DEGs. These findings highlighted potential pathogenic molecules, such as CutC, Gria3, Rbfox3, and others related to Xpo7 haploinsufficiency.

As its name suggests, the basic function of Xpo7 is nuclear export. In fact, among the 45 molecules that we found as Xpo7 direct-effect molecules, 33 were upregulated in the of the frontal cortex cell nuclei of Xpo7$^{+/-}$ mice, indicating that these molecules are affected by the nuclear export deficiency of Xpo7 haploinsufficiency (Fig. 2F). However, CutC, identified as the molecule that was most affected by Xpo7 deficiency in Xpo7$^{+/-}$ mice, was reduced in the nucleus, thereby indicating that Xpo7 plays a role in the nuclear import of CutC (Fig. 2B–D). Previous studies have also reported that Xpo7 functions both as an exportin and an importin (Aksu et al, 2018). Our results further show that Xpo7 has both functions of nuclear export and import.

The 45 molecules identified as Xpo7 direct-effect molecules have inspired several hypotheses about their involvement in pathogenesis. One of the most significantly affected molecules in Xpo7$^{+/-}$ mice is CutC. Although a prior study suggests CutC as a binding target of Xpo7, findings from a study using anti-Xpo7 nanobodies that inhibit Xpo7 indicate no discernible impact on CutC nuclear transport (Aksu et al, 2018). However, CutC was absent from the nucleus in Xpo7$^{-/-}$ mice and demonstrated an ~50% reduction in Xpo7$^{+/-}$ mice (Fig. 2D). CutC is known for its role in mitigating the cytotoxicity of copper. In CutC-knocked-down HepG2 cells, copper levels remained unaltered, but apoptosis increased (Kunjunni et al, 2016). Additionally, further addition of copper loading to the culture medium exacerbated apoptosis. In Caenorhabditis elegans, a 75% knockdown of CutC resulted in organ-wide damage (Calafato et al, 2008). Decreased protection against damage from copper may be similar to a condition of increased copper toxicity. A typical disease with copper toxicity is Wilson's disease, in which patients present with neurological symptoms with reported cases of psychosis (An et al, 2022; Grover et al, 2014). Excess copper causes oxidative stress (An et al, 2022). Because the pathogenetic effect of oxidative stress is associated with a variety of neurological disorders, this may also be related to schizophrenia (Cuenod et al, 2022). Furthermore, because copper concentrations are physiologically high in the locus coeruleus and substantia nigra, these regions may be affected by a reduction of CutC and reduced

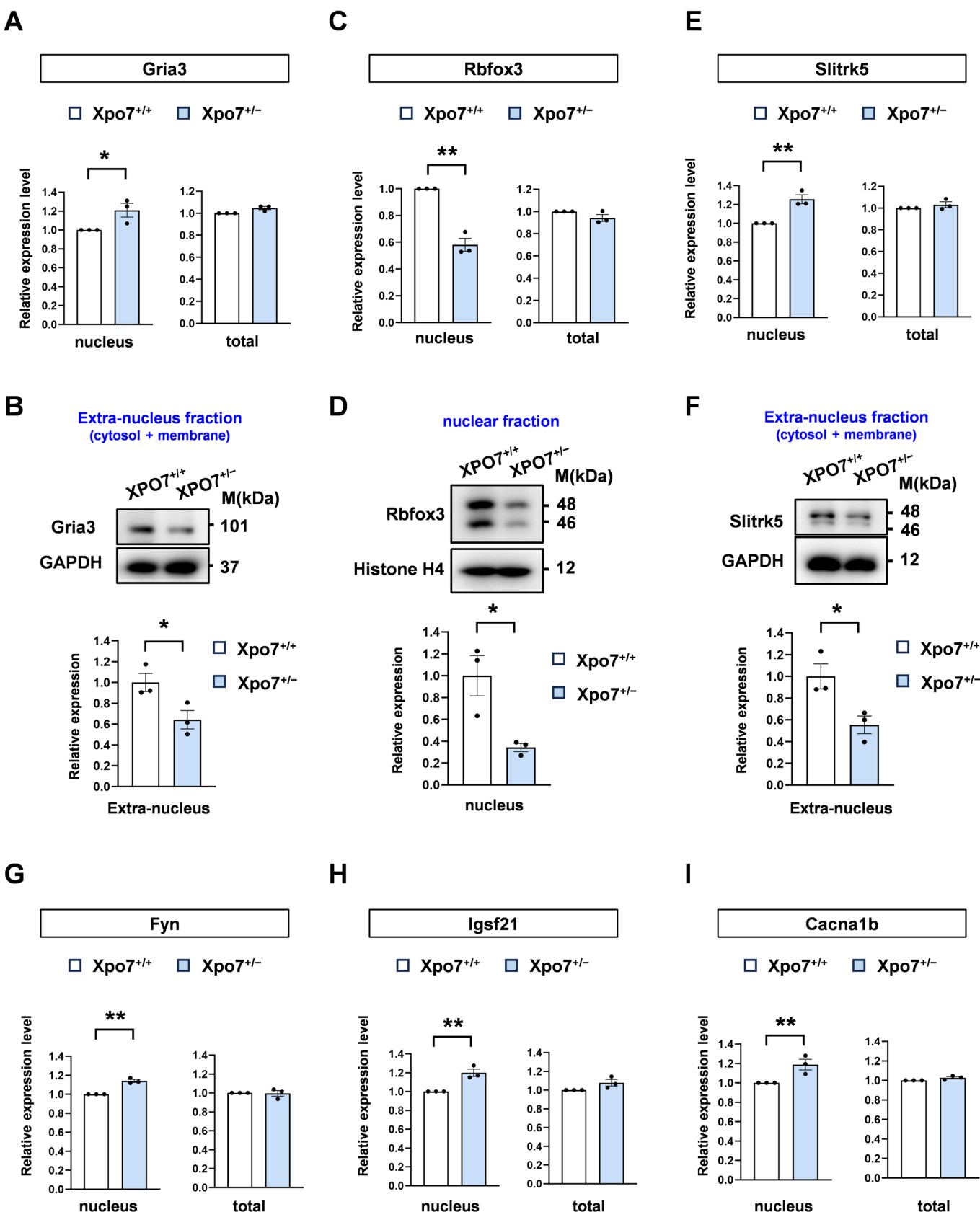

**Figure 3.   Detailed analysis of Xpo7-binding and Xpo7 haploinsufficiency-affected molecules.**

(A) Relative expression of Gria3 in nuclei and whole cells from the frontal cortex of Xpo7$^{+/+}$ and Xpo7$^{+/-}$ mice by LC-MS/MS analysis. *$P < 0.05$ ($P = 0.044$; $N = 3$ experiments, three mice per group/experiment; Tukey's HSD test). As described in Figs. 1C and 2C, three Xpo7$^{+/+}$ mice and three Xpo7$^{+/-}$ mice at 1 month of age were combined as one sample each to obtain the relative change of Xpo7$^{+/-}$ mice to Xpo7$^{+/+}$ mice. Similarly, three Xpo7$^{+/+}$ mice and three Xpo7$^{+/-}$ mice at 3 and 6 months of age were also analyzed to determine the relative change of Xpo7$^{+/-}$ mice to Xpo7$^{+/+}$ mice. Consequently, the relative change of Xpo7$^{+/-}$ mice to Xpo7$^{+/+}$ mice at 1, 3, and 6 months were statistically analyzed to identify the molecules that change consistently in Xpo7$^{+/-}$ mice regardless of their age in months. Data are expressed as the mean ± s.e.m. (B) Western blot analysis of Gria3 in the extranuclear fraction from the frontal cortex of Xpo7$^{+/+}$ and Xpo7$^{+/-}$ mice at 3 months of age. *$P < 0.05$ ($P = 0.044$; $N = 3$ experiments; Tukey's HSD test). Data are expressed as the mean ± s.e.m. (C) Relative expression of Rbfox3 in nuclei and whole cells from the frontal cortex of Xpo7$^{+/+}$ and Xpo7$^{+/-}$ mice by LC-MS/MS analysis. **$P < 0.01$ ($P = 0.00095$; $N = 3$ experiments, three mice per group/experiment; Tukey's HSD test). As described in (A), the LC-MS/MS analysis data were derived from pairs of mice at 1, 3, and 6 months of age. Data are expressed as the mean ± s.e.m. (D) Western blot analysis of Rbfox3 in the nuclear fraction from the frontal cortex of Xpo7$^{+/+}$ and Xpo7$^{+/-}$ mice at 3 months of age. *$P < 0.05$ ($P = 0.025$; $N = 3$ experiments; Tukey's HSD test). Data are expressed as the mean ± s.e.m. (E) Relative expression of Slitrk5 in nuclei and whole cells from the frontal cortex of Xpo7$^{+/+}$ and Xpo7$^{+/-}$ mice by LC-MS/MS analysis. **$P < 0.01$ ($P = 0.0057$; $N = 3$ experiments, three mice per group/experiment; Tukey's HSD test). As described in (A), the LC-MS/MS analysis data were derived from pairs of mice at 1, 3, and 6 months of age. Data are expressed as the mean ± s.e.m. (F) Western blot analysis of Slitrk5 in the extranuclear fraction from the frontal cortex of Xpo7$^{+/+}$ and Xpo7$^{+/-}$ mice at 3 months of age. *$P < 0.05$ ($P = 0.034$; $N = 3$ experiments; Tukey's HSD test). Data are expressed as the mean ± s.e.m. (G-I) Relative expression of Fyn, Igsf21, and Cacna1b in nuclei and whole cells from the frontal cortex of Xpo7$^{+/+}$ and Xpo7$^{+/-}$ mice by LC-MS/MS analysis. **$P < 0.01$ (Fyn $P = 0.034$; Igsf21 $P = 0.0069$; Cacna1b $P = 0.026$; $N = 3$ experiments, three mice per group/experiment; Tukey's HSD test). Data are expressed as the mean ± s.e.m. As described in (A), the LC-MS/MS analysis data were derived from pairs of mice at 1, 3, and 6 months of age. Source data are available online for this figure.

protection against damage from copper (Rihel, 2018; Szerdahelyi and Kása, 1986). Further investigations are necessary to determine the effect of halving CutC in the nucleus of frontal cortex cells of Xpo7$^{+/-}$ mice. Copper-induced damage may be a candidate for pathogenesis in Xpo7 haploinsufficiency.

The involvement of Gria3 in Xpo7-mediated pathological pathways is significant in its impact on schizophrenia-related pathology. In Xpo7$^{+/-}$ mice, the expression of Gria3 outside the cell nucleus is reduced by half, potentially resembling the effects observed with PTVs of one allele of GRIA3. These PTVs have been linked to a 20.1-fold increase in the odds ratio of schizophrenia onset (Singh et al, 2022). Given its gene location on the X chromosome, gender effects related to GRIA3 PTVs are essential. Numerous epilepsy cases associated with GRIA3 mutations have been reported, with a higher occurrence among males (Trivisano et al, 2020). Male mice carrying Gria3 mutations show increased aggression, whereas female mice do not (Adamczyk et al, 2012). Although explicit sex-based differences have not been documented in patients with schizophrenia with GRIA3 mutations, if females carrying the GRIA3 mutation face a similar schizophrenia risk as males, the reduced extranuclear presence of Gria3 due to Xpo7 mutations may also pose a pathogenesis for schizophrenia. In addition, an electrophysiological study is essential in the future to further elucidate the effect of the reduction of the extranuclear Gria3 level in Xpo7 haploinsufficiency pathogenesis.

The reduction of Rbfox3 in the nucleus is also a potential pathogenic factor. Rbfox3, also known as NeuN, plays a role in miRNA biogenesis, neuronal differentiation, and synapse formation (Kim et al, 2013; Kim et al, 2014; Lin et al, 2016). Mutations in Rbfox3 have been reported in autism spectrum disorder (ASD), intellectual disability, and epilepsy (Cooper et al, 2011; Lal et al, 2013; Utami et al, 2014). Previous investigations into mutant mice of Rbfox3 were conducted using knockout mice, so the pathogenic potential of Rbfox3 haploinsufficiency is unclear (Lin et al, 2016). As Rbfox3 is primarily localized within the nucleus, halving its expression in the nucleus of Xpo7$^{+/-}$ mice corresponds to Rbfox3 haploinsufficiency (Fig. 3C). Consequently, further research into Rbfox3 haploinsufficiency is necessary to elucidate its effects. However, considering the diverse neuronal functions attributed to

Rbfox3, it may also contribute to the pathogenesis of Xpo7 haploinsufficiency.

The behavioral analysis showed that Xpo7$^{+/-}$ mice displayed cognitive deficits and impaired pre-pulse inhibition from 3 to 6 months of age. Additionally, deficits in social novelty preference appeared significantly at 6 months but were not obvious at 3 months. This phenotype resembles the gradual onset of symptoms, such as social deficits, seen in schizophrenia patients. It is necessary to conduct a wider range of behavioral tests in the future to confirm whether further progressive abnormal behaviors exist in in Xpo7$^{+/-}$ mice. The potential molecular mechanisms underlying this progressive pathology in Xpo7$^{+/-}$ mice are complex and may involve the persistent accumulation of cellular pathology linked to CutC, Gria3, and Rbfox3 abnormalities. Furthermore, we identified DEGs related to Xpo7 haploinsufficiency pathology, particularly those associated with progressive pathologies. For example, Mt-Atp6 is the causative gene of mitochondrial diseases such as NARP syndrome and Leigh syndrome, characterized by neuropsychiatric symptoms (Stendel et al, 2020). Gpm6a is involved in neuronal polarity and is downregulated in depression (Fuchsova et al, 2015; Honda et al, 2017). Elevated expression of Ubash3b may indicate dopamine pathology, while increased Slc9a9 expression has been linked to ASD (Patak et al, 2017; Savell et al, 2020). Additionally, synapse-related and regulatory molecules such as Gabra2, Homer1, and Mef2c were included in the DEGs (Harrington et al, 2016). The integration of these pathologies may contribute to a progressive pathology, highlighting the importance of focusing on progressive molecular mechanisms in this study. These changes in gene expression and nuclear translocation of molecules may be associated with the changes in the morphology of neurons. We observed a significant decrease in the apical and basal dendrites of L2/3 neurons and the apical dendrites of L5/6 neurons in the frontal cortex in Xpo7$^{+/-}$ mice (Fig. EV2A,B). Although there were no significant differences in spine density per dendrite length by Golgi staining analysis, a decrease in dendrites suggests a total decrease of synapses in Xpo7$^{+/-}$ mice. Further analyses are necessary to determine the synaptic pathologies of Xpo7$^{+/-}$ mice. In this study, we focused on analyzing neurons; however, the glial cell pathology, including the pathology of white matter, is also

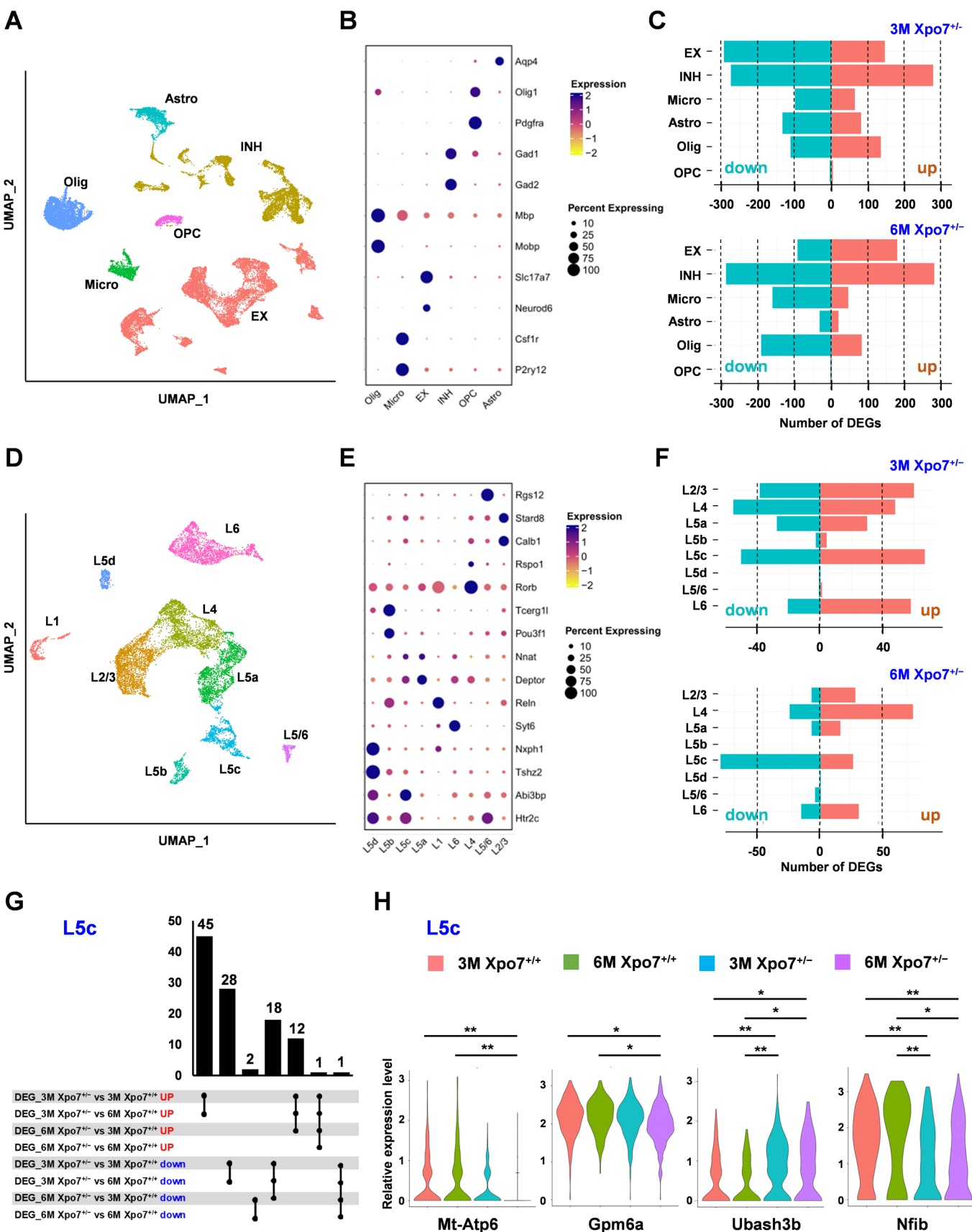

Figure 4.  Cell-type-specific transcriptional analysis of excitatory neurons in the frontal cortex of Xpo7$^{+/-}$ mice.

(A) Uniform manifold approximation and projection (UMAP) plot of cells from the frontal cortex and striatum. (B) Dot plot of molecular markers of cells in (A). Astro astrocytes, EX excitatory neurons, INH inhibitory neurons, Micro microglia, Olig oligodendrocytes, OPC oligodendrocyte precursor cell. (C) Bar graph of the number of DEGs in different frontal cortex and striatum cell populations in Xpo7$^{+/-}$ mice at 3 and 6 months of age. DEGs are |log2 fold change| > 0.25, $P < 0.05$. DEGs were computed using MAST method in the Seurat software. (D) UMAP plot of the excitatory neuron subtypes in (A). (E) Dot plot of molecular markers in excitatory neuron subtypes in the frontal cortex. (F) Bar graph of the number of DEGs in excitatory neuron subtypes in the frontal cortex in Xpo7$^{+/-}$ mice at 3 and 6 months of age. DEGs are |log2 fold change| > 0.25, $P < 0.05$. DEGs were computed using MAST method in the Seurat software. (G) Bar graph of the number of DEGs in L5c neuron subtypes in the frontal cortex in Xpo7$^{+/-}$ mice at 3 and 6 months of age. The linked DEGs below the bar graph indicate that the DEGs corresponding to each bar belong to all the indicated rows. For example, 45 DEGs belong to both "DEGs 3 M Xpo7 + /− vs 3 M Xpo7 + /+ up" and "DEGs 3 M Xpo7 + /− vs 3 M Xpo7 + /+ up". DEGs are |log2 fold change| > 0.25, $P < 0.05$. DEGs were computed using MAST method in the Seurat software. (H) Relative mRNA expression levels of Mt-Atp6, Gpm6a, Ubash3b, and Nfib. *$P < 0.05$, **$P < 0.01$ (Mt-Atp6 6 M Xpo7$^{+/-}$ vs 3 M Xpo7$^{+/+}$ $P = 0.00059$; Mt-Atp6 6 M Xpo7$^{+/-}$ vs 6 M Xpo7$^{+/+}$ $P = 0.0028$; Gpm6a 6 M Xpo7$^{+/-}$ vs 3 M Xpo7$^{+/+}$ $P = 0.023$; Gpm6a 6 M Xpo7$^{+/-}$ vs 6 M Xpo7$^{+/+}$ $P = 0.041$; Ubash3b 6 M Xpo7$^{+/-}$ vs 3 M Xpo7$^{+/+}$ $P = 0.013$; Ubash3b 6 M Xpo7$^{+/-}$ vs 6 M Xpo7$^{+/+}$ $p = 0.010$; Ubash3b 3 M Xpo7$^{+/-}$ vs 3 M Xpo7$^{+/+}$ $P = 0.00013$; Ubash3b 3 M Xpo7$^{+/-}$ vs 6 M Xpo7$^{+/+}$ $P = 0.00019$; Nfib 6 M Xpo7$^{+/-}$ vs 3 M Xpo7$^{+/+}$ $p = 0.00025$; Nfib 6 M Xpo7$^{+/-}$ vs 6 M Xpo7$^{+/+}$ $P = 0.030$; Nfib 3 M Xpo7$^{+/-}$ vs 3 M Xpo7$^{+/+}$ $p = 0.000021$; Nfib 3 M Xpo7$^{+/-}$ vs 6 M Xpo7$^{+/+}$ $P = 0.0032$; 3 M Xpo7$^{+/+}$ $N = 269$ cells, 6 M Xpo7$^{+/+}$ $N = 141$ cells, 3 M Xpo7$^{+/-}$ $N = 288$ cells, 6 M Xpo7$^{+/-}$ $N = 297$ cells; Wilcoxon rank-sum test). Source data are available online for this figure.

important to schizophrenia pathology. Detailed analysis of glial cells and white matter together with axons will be the focus of future studies.

Previous research findings on Xpo7 are also crucial for the future pathological analysis of Xpo7 haploinsufficiency. Previous studies have examined Xpo7 in proliferating cells regarding erythropoiesis and cellular senescence (Innes et al, 2021; Li et al, 2023; Modepalli et al, 2022). For example, Xpo7 knockout resulted in a decrease in Smad3 expression in erythropoiesis (Modepalli et al, 2022). In addition, in proliferating cells, suppressing Xpo7 leads to senescence inhibition through the involvement of E2A/TCF3 and p21cip1 (Innes et al, 2021). However, Smad3 levels did not decrease in the brains of Xpo7$^{-/-}$ mice and E2A/TCF3 and p21cip1 were not detected in our proteome analysis. Therefore, it is unclear whether reducing Xpo7 affects neuronal senescence, as neurons are non-dividing cells. The suppression of neuronal senescence in the nervous system may be influenced by decreased CutC or Rbfox3, which may promote senescence in neurons because their reduction leads to increased cellular damage or dysregulation of multiple cellular functions. Similarly, previous studies identified Hat1 and Nampt as target molecules impacted by Xpo7 inhibition (Aksu et al, 2018). However, Hat1 was not detected in our proteome analysis. Additional studies are needed to elucidate the effect of Hat1 in Xpo7 haploinsufficiency. Nampt exhibited reduced nuclear expression levels in Xpo$^{-/-}$ cells; however, no significant reductions were observed in Xpo7$^{+/-}$ mice. However, our single-nucleus RNA sequencing analysis revealed Nampt as one of the identified DEGs in Xpo7$^{+/-}$ mice. Thus, future research into these molecules may uncover additional Xpo7 nervous system pathologies.

Moreover, the clinical information for patients with schizophrenia with XPO7 variants, including detailed case reports, is important for further understanding the pathogenesis of XPO7 deficiency. We hypothesize the presence of a progressive molecular pathogenesis associated with long DUP and relapse, whereas large-scale analyses of patients with schizophrenia reveal evidence of cognitive impairment at the onset, but this impairment is stable throughout the course of the disease with appropriate treatment (McCutcheon et al, 2023; Velthorst et al, 2021). The treatment responsiveness as well as the cognitive and social course of patients with XPO7 variants are necessary for further interpretation of our findings.

There are multiple ways to apply this research to clinical practice. Firstly, what is the appropriate treatment for XPO7 haploinsufficiency? One approach for treating XPO7 haploinsufficiency is to restore the expression of XPO7 itself; however, further studies are needed to verify whether this approach can be used as a treatment. For example, determining whether behavioral abnormalities are present when Xpo7 expression is reduced only after birth will make it clear to what extent the Xpo7 haploinsufficiency results in pathogenesis after birth. Examining whether recovery is possible by restoring Xpo7 expression after birth or after the onset of the disease (after the age of 3 months) in Xpo7 haploinsufficiency mice will make it clear whether restoring the expression of Xpo7 is a viable treatment. In addition, the latter experiments reveal to what extent Xpo7 haploinsufficiency in embryogenesis can form behavioral and molecular pathologies. In addition, it is crucial to consider the broader implications for schizophrenia, which are caused by a complex interaction of common genetic variants rather than XPO7 mutations. Even though rare variants like XPO7 insufficiency may display specific pathological mechanisms, they are also likely to involve pathogenic pathways relevant to common schizophrenia variants. Therefore, it is essential to investigate whether the candidate pathologies identified in Xpo7 insufficiency are also involved in other rare but high-odds haploinsufficiency variants, as such a common pathogenic pathway would be shared with a broader spectrum of schizophrenia patients. In this context, the discovery of abnormalities in Gria3 in Xpo7$^{+/-}$ mice and disease-associated DEGs such as Gria3, Grin2A, Herc1, and Trio is significant. It is crucial to understand how these rare variants of high-risk gene products interact with each other to elucidate the molecular mechanisms underlying the onset and progression of schizophrenia. This understanding may provide insights into preventing the onset and progression of the disease in a broader range of patients with schizophrenia, not just those with XPO7 haploinsufficiency.

In conclusion, our investigation of Xpo7$^{+/-}$ mice revealed the target molecules involved in nuclear trafficking by Xpo7 in the nervous system and the pathophysiological changes in the expression of cell groups and associated DEGs. It is crucial to identify shared pathology among various schizophrenia-associated animal models, including this study's findings, for a better understanding of the broader pathology of schizophrenia.

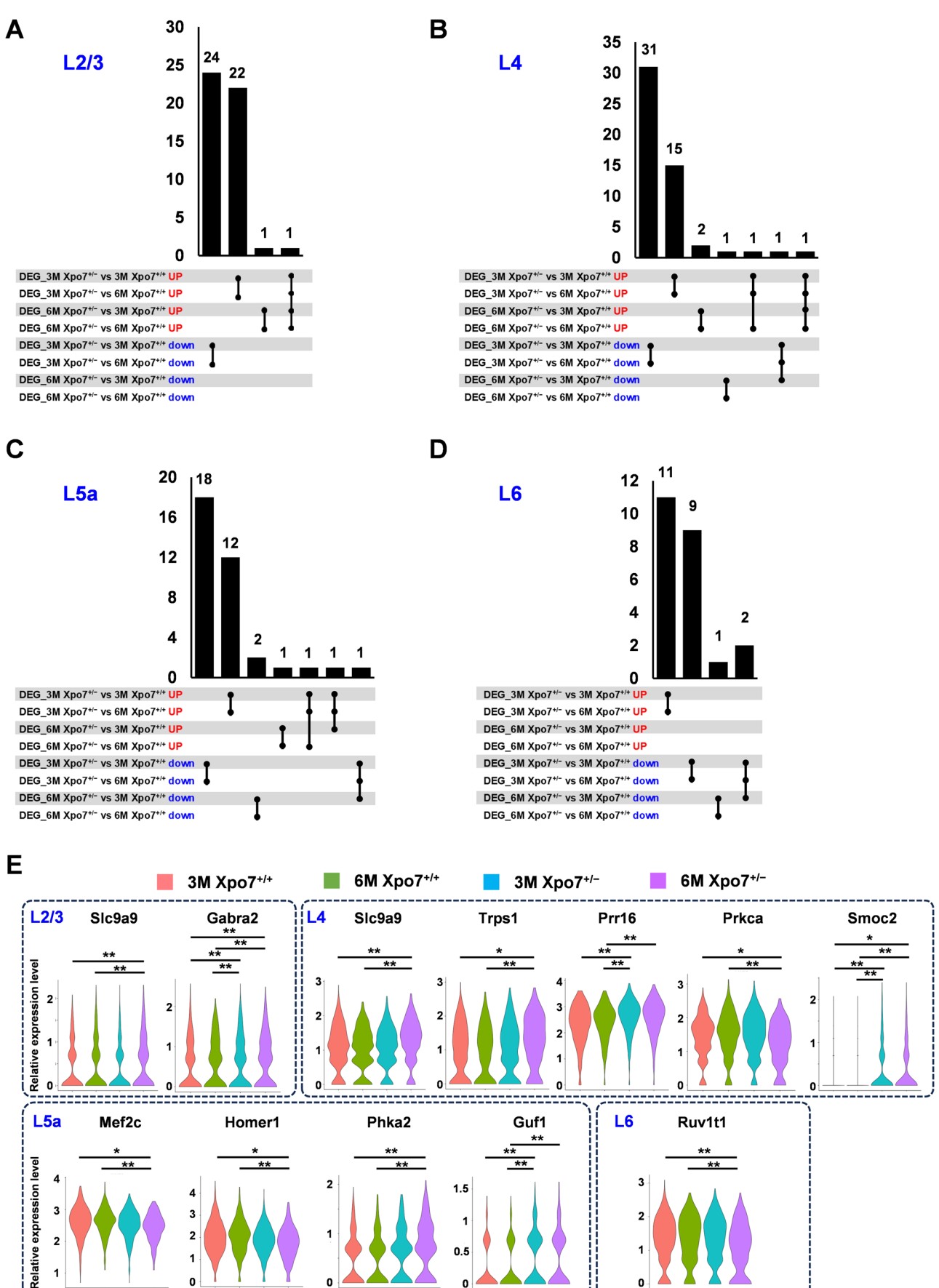

◀ **Figure 5.  Cell-type-specific transcriptional analysis of excitatory neurons in each layer of the frontal cortex of Xpo7$^{+/-}$ mice.**

(A–D) Bar graph of the number of DEGs in Layers 2/3, 4, 5a, and 6 (L2/3, L4, L5a, L6) neuron subtypes in the frontal cortex of Xpo7$^{+/-}$ mice at 3 and 6 months of age. The linked DEGs below the bar graph indicate that the DEGs corresponding to each bar belong to all the indicated rows. DEGs were computed using MAST method in the Seurat software. (E) Relative mRNA expression levels of Slc9a9 and Gabra2 in L2/3 neurons; Slc9a9, Trps1, Prr16, and Prkca in L4 neurons; Mef2c, Homer1, Phka2, and Guf1 in L5a neurons; and Runx1t1 in L6 neurons. *$P < 0.05$, **$P < 0.01$ (L2/3 Slc9a9 6 M Xpo7$^{+/-}$ vs 3 M Xpo7$^{+/+}$ $P = 0.0049$; L2/3 Slc9a9 6 M Xpo7$^{+/-}$ vs 6 M Xpo7$^{+/+}$ $P = 0.0027$; L2/3 Gabra2 6 M Xpo7$^{+/-}$ vs 3 M Xpo7$^{+/+}$ $P = 0.0040$; L2/3 Gabra2 6 M Xpo7$^{+/-}$ vs 6 M Xpo7$^{+/+}$ $P = 2.59E-07$; L2/3 Gabra2 3 M Xpo7$^{+/-}$ vs 3 M Xpo7$^{+/+}$ $P = 2.33E-12$; L2/3 Gabra2 3 M Xpo7$^{+/-}$ vs 6 M Xpo7$^{+/+}$ $P = 3.40E-19$; L4 Slc9a9 6 M Xpo7$^{+/-}$ vs 3 M Xpo7$^{+/+}$ $P = 0.0067$; L4 Slc9a9 6 M Xpo7$^{+/-}$ vs 6 M Xpo7$^{+/+}$ $P = 9.17E-05$; L4 Trps1 6 M Xpo7$^{+/-}$ vs 3 M Xpo7$^{+/+}$ $P = 0.015$; L4 Trps1 6 M Xpo7$^{+/-}$ vs 6 M Xpo7$^{+/+}$ $P = 0.00060$; L4 Prr16 6 M Xpo7$^{+/-}$ vs 3 M Xpo7$^{+/+}$ $P = 0.00056$; L4 Prr16 3 M Xpo7$^{+/-}$ vs 3 M Xpo7$^{+/+}$ $P = 7.33E-10$; L4 Prr16 3 M Xpo7$^{+/-}$ vs 6 M Xpo7$^{+/+}$ $P = 3.25E-06$; L4 Prkca 6 M Xpo7$^{+/-}$ vs 3 M Xpo7$^{+/+}$ $P = 0.038$; L4 Prkca 6 M Xpo7$^{+/-}$ vs 6 M Xpo7$^{+/+}$ $P = 0.00039$; L4 Smoc2 6 M Xpo7$^{+/-}$ vs 3 M Xpo7$^{+/+}$ $P = 0.011$; L4 Smoc2 6 M Xpo7$^{+/-}$ vs 6 M Xpo7$^{+/+}$ $P = 3.61E-07$; L4 Smoc2 3 M Xpo7$^{+/-}$ vs 3 M Xpo7$^{+/+}$ $P = 1.58E-07$; L4 Smoc2 3 M Xpo7$^{+/-}$ vs 6 M Xpo7$^{+/+}$ $P = 2.70E-13$; L5 Mef2c 6 M Xpo7$^{+/-}$ vs 3 M Xpo7$^{+/+}$ $P = 2.40E-05$; L5 Mef2c 6 M Xpo7$^{+/-}$ vs 6 M Xpo7$^{+/+}$ $P = 0.0032$; L5 Homer1 6 M Xpo7$^{+/-}$ vs 3 M Xpo7$^{+/+}$ $P = 0.042$; L5 Homer1 6 M Xpo7$^{+/-}$ vs 6 M Xpo7$^{+/+}$ $P = 0.0012$; L5 Phka2 6 M Xpo7$^{+/-}$ vs 3 M Xpo7$^{+/+}$ $P = 0.0018$; L5 Phka2 6 M Xpo7$^{+/-}$ vs 6 M Xpo7$^{+/+}$ $P = 0.0087$; L5 Guf1 6 M Xpo7$^{+/-}$ vs 3 M Xpo7$^{+/+}$ $P = 0.0070$; L5 Guf1 3 M Xpo7$^{+/-}$ vs 3 M Xpo7$^{+/+}$ $P = 2.07E-06$; L5 Guf1 3 M Xpo7$^{+/-}$ vs 6 M Xpo7$^{+/+}$ $P = 0.0031$; L6 Runx1t1 6 M Xpo7$^{+/-}$ vs 3 M Xpo7$^{+/+}$ $P = 0.00027$; L6 Runx1t1 6 M Xpo7$^{+/-}$ vs 6 M Xpo7$^{+/+}$ $P = 0.013$; 3 M Xpo7$^{+/+}$ $N = 688$ cells (L2/3), 578 cells (L4), 583 cells (L5a), 750 cells (L6); 6 M Xpo7$^{+/+}$ $N = 641$ cells (L2/3), 522 cells (L4), 398 cells (L5a), 435 cells (L6); 3 M Xpo7$^{+/-}$ $N = 712$ cells (L2/3), 697 cells (L4), 493 cells (L5a), 595 cells (L6), 6 M Xpo7$^{+/-}$ $N = 423$ cells (L2/3), 334 cells (L4), 336 cells (L5a), 420 cells (L6); Wilcoxon rank-sum test). Source data are available online for this figure.

# Methods

**Reagents and tools table**

| Reagent/resource | Reference or source | Identifier or catalog number |
|---|---|---|
| **Experimental models** | | |
| Xpo7 knock out mice (Xpo7$^{+/-}$ mice) | This study | N/A |
| C57BL/6J mice | CLEA Japan | RRID:IMSR_JCL:JCL:mIN-0003 |
| **Antibodies** | | |
| Anti-XPO7 (A-11) | Santa Cruz Biotechnology | sc-390025 |
| Anti-CutC (E-1) | Santa Cruz Biotechnology | sc-515505 |
| Anti-Histone H4 | abcam | ab31830 |
| Anti-Gria3 | Synaptic Systems | 182 203 |
| Anti-Rbfox3 | abcam | ab104224 |
| Anti-Slitrk5 | Proteintech | 21496-1-AP |
| Anti-Igsf21 | Proteintech | 21465-1-AP |
| Anti-GAPDH | Merck Millipore | MAB374 |
| Anti-Cux1 | Proteintech | 11733-1-AP |
| Anti-Tbr1 | abcam | ab183032 |
| HRP-linked anti-rabbit IgG | GE Healthcare | NA934 |
| HRP-linked anti-mouse IgG | GE Healthcare | NA931 |
| Cy3-conjugated anti-rabbit IgG | Jackson Laboratory | 711-165-152 |
| **Oligonucleotides and other sequence-based reagents** | | |
| crRNA 5′-AGGGUGUUUGAUCCAAGUGCguuuuagagcuaugcuguuuug-3′ | This study | N/A |
| crRNA 5′-CUUUUGGCUUCUAGGGUACCguuuuagagcuaugcuguuuug-3′ | This study | N/A |
| Xpo7 genotyping Forward Primer 5′-GCCACACCCCATATTGCTTATG-3′ | This study | N/A |
| Xpo7 genotyping Reverse Primer 5′-GGAACCACAGGTTAAGTGACAC-3′ | This study | N/A |
| **Chemicals, enzymes, and other reagents** | | |
| ECL Prime Western Blotting Detection Reagent | GE Healthcare | RPN2232 |
| SliceGolgi kit | BioEnno | 003760 |
| G–Sepharose beads | GE Healthcare | 17061801 |

| Reagent/resource | Reference or source | Identifier or catalog number |
|---|---|---|
| Nuclei Extraction Buffer | Miltenyi | 130-128-024 |
| Debris Removal Solution | Miltenyi | 130-109-398 |
| Trypsin/Lys-C Mix | Promega | V5071 |
| GL-Tip SDB | GL Sciences | 7820-11200 |
| lauryl maltose neopentyl glycol | Anatrace | NG310 |
| decyl maltose neopentyl glycol | Anatrace | NG322 |
| Chromium Next GEM Single Cell Multiome ATAC + Gene Expression Reagent Bundle | 10x Genomics. | PN-1000285 |
| protease inhibitor cocktail | Merck | 539134-1SETCN |
| **Software** | | |
| DIA-NN version 1.8.1 | Demichev et al, 2020 | https://github.com/vdemichev/DiaNN |
| Cell Ranger Arc version 2.0.2 | 10x Genomics | https://www.10xgenomics.com/jp/support/software/cell-ranger-arc/latest |
| Seurat version 5.0.1 | | https://satijalab.org/seurat/ |
| GraphPad Prism 8.4.3 | GraphPad Software | https://www.graphpad.com/features |
| Scds version 1.20.0 | | https://www.bioconductor.org/packages/release/bioc/html/scds.html |
| DecontX version 1.20.0 | | https://www.bioconductor.org/packages/release/bioc/html/celda.html |
| SCTransform v2 (version 0.4.1) | Choudhary and Satija, 2022 | https://github.com/satijalab/sctransform |
| Signac version 1.12.0 | Stuart et al, 2021 | https://stuartlab.org/signac/ |
| MACS2 | | https://github.com/macs3-project/MACS |
| chromVAR version 1.24.0 | Schep et al, 2017 | https://greenleaflab.github.io/chromVAR/articles/Introduction.html |
| STRING database | Szklarczyk et al, 2023 | https://string-db.org/ |
| Infomap algorithm | Rosvall and Bergstrom et al, 2008 | https://www.mapequation.org/infomap/ |

## Ethics

This study followed the Guidelines for Proper Conduct of Animal Experiments by the Science Council of Japan, and the Helsinki Declaration. The protocol was approved by the Committee on Animal Experiments and Gene Recombination Experiments of the Tokyo Medical and Dental University (G2020-002A and A2020-113A).

## Mice

Littermates of $Xpo7^{+/+}$, $Xpo7^{+/-}$, and $Xpo7^{-/-}$ mice were housed in standard cages in a temperature- and humidity-controlled room with a 12-h light/dark cycle (lights on at 08:00). Investigators were blind to the treatment of mice during experimentation and data analysis. Male mice with the C57BL/6J background were used for conducting all behavioral tests and biochemical analyses. The generated $Xpo7^{+/-}$ mice were backcrossed more than five times with C57BL6/J mice prior to the analysis.

## Behavioral tests

All behavioral tests were analyzed using a video-computerized tracking system (SMART, Panlab, Barcelona, Spain).

### Open-field test

Mice were placed in an open-field box ($40 \times 40 \times 22$ cm) and allowed to explore freely for 10 min. The total distance moved and the time spent in the central zone ($20 \times 20$ cm) were measured.

### Three-chamber sociability test

Mice were situated within a three-chambered box, each chamber measuring $40 \times 20 \times 22$ cm (L, W, H), with dividers featuring inter-chamber access openings. The test comprised three sessions: habituation, sociability, and social novelty preference. During the 5-minute habituation period, the mice were allowed to explore; then, they were confined in the central chamber for 5 min. During the sociability portion of the experiment, an unfamiliar mouse was placed in a wire cup within one side chamber. The test mouse was given 10 min to explore all chambers freely. In the social novelty preference session, a new, unfamiliar mouse was placed in the wire cup on the opposite side chamber. Once again, the test mouse had 10 min for exploration. Time spent in each chamber and actual interaction time were recorded. Figure interaction times are based on actual interaction durations.

### Elevated plus maze test

The elevated plus maze consisted of two open and two closed arms (each $30 \times 6$ cm, with 15 cm high walls), elevated 50 cm above the

floor. The mice were placed in the central square of the maze, and their activity was recorded for 5 min. The time spent in the open and closed arms was measured.

### PPI test

The experiment utilized sound-attenuating startle boxes (Panlab, Barcelona, Spain). After a 5-minute acclimation period with background noise set at 65 dB, each mouse was exposed to 10 blocks of six startle stimuli presented in a pseudorandomized order. The trial types included startle-only (40 ms, 120 dB sound burst) and five prepulse trials (120 dB startle stimulus preceded by a 20 ms prepulse at 69, 73, 77, 81, or 85 dB, 100 ms earlier). The maximum startle response was recorded for each stimulus.

### Y maze test

Mice were placed at the end of one arm of the maze and then allowed to explore for 8 min. The percentage of spontaneous alternations (the alternation rate) was calculated by dividing the number of entries into a new arm, different from the previous one, by the total number of arm entries.

### Novel object recognition test

The training session used two identical objects, and the mice were allowed a 10-minute exploration period. Immediately following this period, one of the objects was replaced with a novel object, and the mice were allowed to examine it again until they reached a total exploration time of 30 s. Omitting a break between sessions in the novel object recognition test aims to maintain a consistency in test structure with the three-chamber test.

## Generation of Xpo7 knockout mice

Xpo7 knockout mice were generated using a cloning-free CRISPR/Cas system described previously (Aida et al, 2015). Both crRNAs and tracrRNA were chemically synthesized and purified by high-pressure liquid chromatography (FASMAC, Japan). The sequences of the crRNA are 5′-AGGGUGUUUGAUCCAAGUGCguuuuagag-cuaugcuguuuug-3′ and 5′-CUUUUGGCUUCUAGGGUACCguuu uagagcuaugcuguuuug-3′. Genotyping was performed using the following primers: 5′-GCCACACCCCATATTGCTTATG-3′ and 5′-GGAACCACAGGTTAAGTGACAC-3′. The generated Xpo7$^{+/-}$ mice were backcrossed more than five times with C57BL6/J mice prior to the analysis.

## Genotyping polymerase chain reaction (PCR) of Xpo7 knockout mice

The primer, template, and enzyme used were as follows. Forward Primer: 5′-GCCACACCCCATATTGCTTATG-3′, Reverse Primer: 5′-GGAACCACAGGTTAAGTGACAC-3′. Template: DNA extracted from the ear by the alkaline lysis method. Enzyme: TaKaRa Ex Premier™ DNA Polymerase (Takara). The annealing temperature was 57 °C, and the other steps were performed per the manufacturer's protocol of the enzyme.

## Western blot analysis

Samples were lysed in 62.5 mM Tris-HCl (pH 6.8), 2% (w/v) SDS, 2.5% (v/v) 2-mercaptoethanol, 5% (v/v) glycerin, Protease Inhibitor

Cocktail (1:200, 539134-1SETCN, Merck), and 0.0025% (w/v) bromophenol blue. The lysates were separated by SDS-PAGE, transferred onto Immobilon-P Transfer Membranes (Merck Millipore, Burlington, MA, USA) using a semi-dry method, and blocked with 5% milk in TBS/Tween 20 (TBST) (10 mM Tris/Cl, pH 8.0, 150 mM NaCl, and 0.05% Tween 20). Filters were incubated overnight at 4 °C with the following primary antibodies diluted in Can Get Signal solution (Toyobo, Osaka, Japan): anti-XPO7 (A-11) (1:500; sc-390025, Santa Cruz Biotechnology), anti-CutC (E-1) (1:500; sc-515505, Santa Cruz Biotechnology), anti-Histone H4 (1:2000; ab31830,abcam), anti-Gria3 (1:500;182 203, Synaptic Systems), anti-Rbfox3 (1:500; ab104224, abcam), anti-Slitrk5 (1:1000; 21496-1-AP, Proteintech), anti-Igsf21 (1:1000; 21465-1-AP, Proteintech), and anti-GAPDH (1:3000; MAB374, Merck Millipore). Secondary antibodies were HRP-linked anti-rabbit IgG (1:3000; NA934, GE Healthcare, Chicago, IL, USA) and HRP-linked anti-mouse IgG (1:3000; NA931, GE Healthcare, Chicago, IL, USA). Proteins were detected using ECL Prime Western Blotting Detection Reagent (RPN2232, GE Healthcare, Chicago, IL, USA) and a luminescence image analyzer (Image-Quant LAS 500, GE Healthcare, Chicago, IL, USA).

## Immunohistochemistry

Brain samples were fixed with 4% paraformaldehyde and embedded in paraffin. Coronal sections (5-μm thick) were cut using a microtome (Microm HM 335 E, GMI, Ramsey, USA). Immuno-histochemistry was performed using the following primary antibodies, anti-Cux1 (1:200, 11733-1-AP, Proteintech) and anti-Tbr1(1:200, ab183032, abcam). These were detected using Cy3-conjugated anti-rabbit IgG (1:500, 711-165-152, Jackson Laboratory). Nuclei were stained with 4′,6-diamidino-2-phenylindole (0.2 μg/mL in PBS; DOJINDO). Images were acquired under an Olympus FV1200 confocal microscope (Tokyo, Japan).

## Golgi staining and analysis

The mice were deeply anesthetized with ketamine (100 mg/kg, i.p.) and xylazine (10 mg/kg, i.p.) and then perfused with a formalde-hyde/glutaraldehyde fixative (BioEnno, Irvine, CA). After perfusion, the brains were removed, post-fixed overnight in the same fixative, sectioned into 60 μm slices using a vibratome (Leica), and the floating sections were collected in PBS. On-slice Golgi staining was conducted using a SliceGolgi kit (BioEnno, Irvine, CA) based on the manufacturer's protocol, which involved a 7-day Golgi impregnation. After staining, the slices were mounted onto glass slides, dried, dehydrated in 100% ethanol, cleared with xylene, and coverslipped in Permount. Images were captured using an inverted light microscope (BZ-X710, Keyence) with a 10× objective lens for dendritic arborization analysis and a 100× objective lens for spine density and size analysis. Pyramidal neurons were selected from layers 2–3 or layers 5–6 of the frontal cortex, including the cingulate, infralimbic, prelimbic, and dorsal peduncular cortices. The Golgi-stained images were analyzed using ImageJ software (http://rsb.info.nih.gov/ij/). Dendritic arborization was assessed using Sholl analysis with the Simple Neurite Tracer (SNT) plugin (Arshadi et al, 2021). Neurites were traced along the *z* axis for optimal accuracy and the Soma and dendrites were reconstructed using the ROI manager tool. Sholl analysis was performed with the

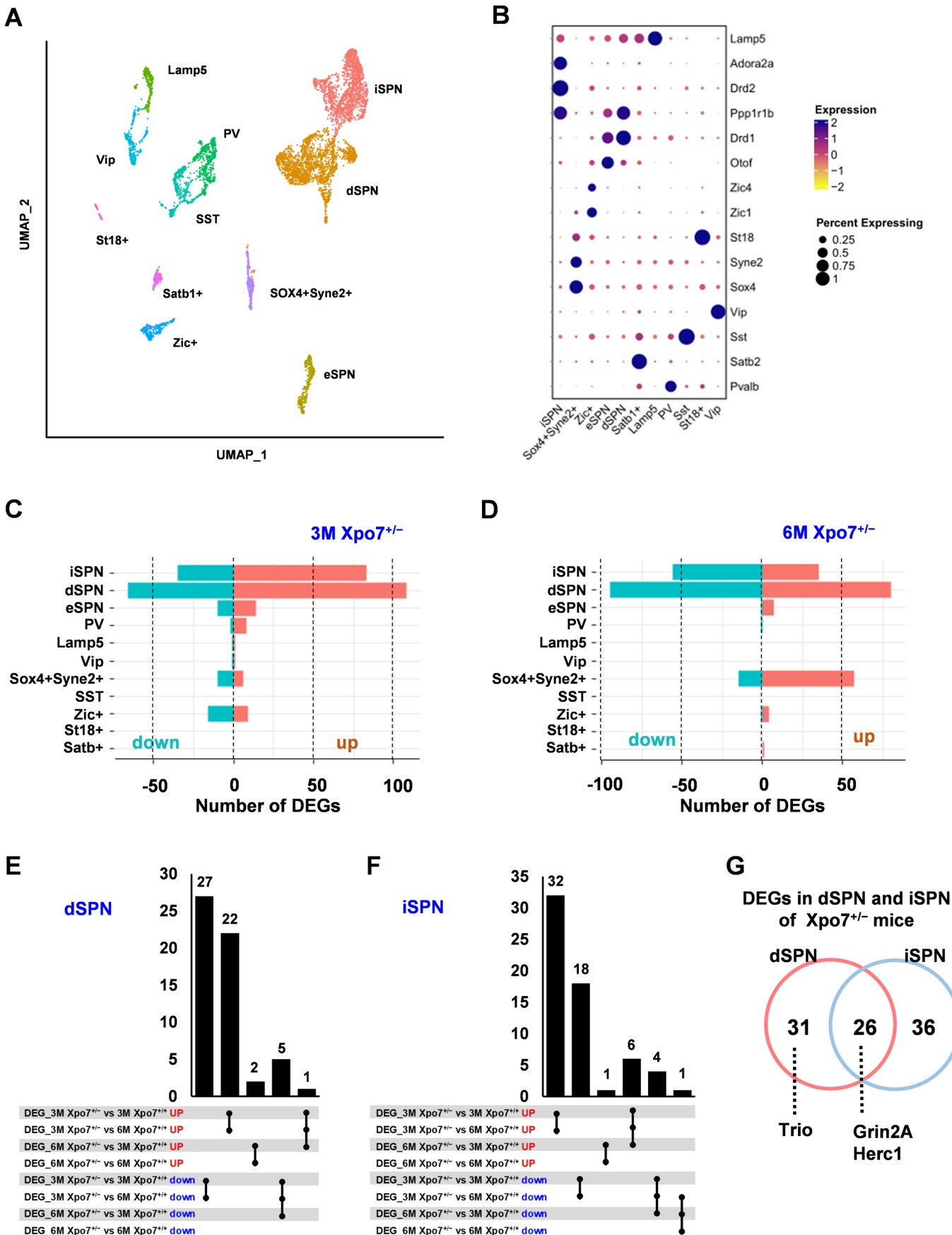

**Figure 6.    Cell-type-specific transcriptional analysis in the striatum of Xpo7^{+/−} mice.**

(A) UMAP plot of inhibitory neuron subtypes from the striatum and frontal cortex. (B) Dot plot of molecular markers in inhibitory neurons from the striatum and frontal cortex. (C) Bar graph of the number of DEGs in inhibitory neurons from the striatum and frontal cortex in Xpo7^{+/−} mice at 3 months of age. DEGs are |log2 fold change| > 0.25, $P < 0.05$. $N = 1021$ cells (dSPN), 906 cells (iSPN), 224 cells (eSPN), 332 cells (PV), 247 cells (Sst), 186 cells (Lamp5), 210 (Vip), 72 cells (clu12), 33 cells (clu14), 154 (clu8), 135 cells (clu9). DEGs were computed using MAST method in the Seurat software. (D) Bar graph of the number of DEGs in inhibitory neurons from the striatum and frontal cortex in Xpo7^{+/−} mice at 6 months of age. DEGs are |log2 fold change| > 0.25, $P < 0.05$. $N = 863$ cells (dSPN), 747 cells (iSPN), 168 cells (eSPN), 233 cells (PV), 160 cells (Sst), 113 cells (Lamp5), 142 cells (Vip), 78 cells (clu12), 28 cells (clu14), 153 cells (clu8), 139 cells (clu9). DEGs were computed using MAST method in the Seurat software. (E) Bar graph of the number of DEGs in dSPN neurons of the striatum in Xpo7^{+/−} mice at 3 and 6 months of age. DEGs at the same age and DEGs at different ages are associated with the pathophysiology of Xpo7 haploinsufficiency. DEGs are |log2 fold change| > 0.25, $P < 0.05$. 3 M Xpo7^{+/+} $N = 540$ cells, 6 M Xpo7^{+/+} $N = 544$ cells, 3 M Xpo7^{+/−} $N = 481$ cells, 6 M Xpo7^{+/−} $N = 319$ cells. DEGs were computed using MAST method in the Seurat software. (F) Bar graph of the number of DEGs in iSPN neurons of the striatum in Xpo7^{+/−} mice at 3 and 6 months of age. DEGs are |log2 fold change| > 0.25, $P < 0.05$. 3 M Xpo7^{+/+} $N = 442$ cells, 6 M Xpo7^{+/+} $N = 487$ cells, 3 M Xpo7^{+/−} $N = 464$ cells, 6 M Xpo7^{+/−} $N = 250$ cells. DEGs were computed using MAST method in the Seurat software. (G) Venn diagrams of (E, F). Grin2A, Herc1, and Trio are included in genes expressing variable dSPN or iSPN in Xpo7^{+/−} mice. Source data are available online for this figure.

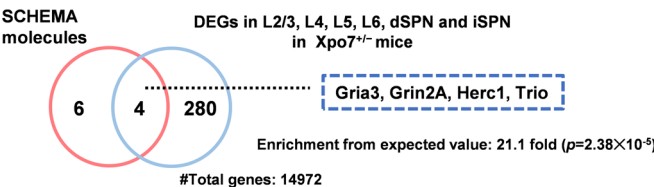

**Figure 7.    DEGs identified through single-nucleus RNA sequencing demonstrate significant overlap with SCHEMA molecules.**

High-risk genes for schizophrenia (SCHEMA molecules) are significantly enriched in DEGs in L2/3, L4, L5, and L6 excitatory neurons and dSPN and iSPN in Xpo7^{+/−} mice. Enrichment from expected value: 21.1-fold ($P = 0.0000238$, Hypergeometric distribution).

shells set at a 10-μm radius and the intersections at each radius were counted. For each animal, 6–12 pyramidal neurons from either layers 2–3 or layers 5–6 were analyzed and the mean number of intersections was used as the representative value. Spine density and size analysis included only pyramidal neurons with clearly identifiable spines and dendrites. Spine density was standardized by the length of the dendrites for comparative analysis. The spine head diameter was measured at the secondary and higher-order dendrites of both the apical and basal dendrites, and defined as the minor axis of an ellipse-approximated spine head. Measurements were taken from 6–14 neurons in each dendrite per animal and the mean spine density and spine head diameter were considered representative values for each animal.

## Immunoprecipitation

Mouse cerebral frontal cortex from Xpo7^{+/+} and Xpo7^{−/−} mice at the age of 1 month were lysed using a homogenizer with lysis buffer (10 mM Tris-HCl, pH 7.5, 150 mM NaCl, 1 mM EDTA, 10% Nonidet P-40, 0.5% protease inhibitor cocktail (1:200, 539134-1SETCN, Merck). The lysates were rotated for 60 min at 4 °C, then centrifuged (16,000 × *g* for 10 min at 4 °C), then incubated with 1 μg of anti-XPO7 antibodies (A-11: sc-390025, Santa Cruz Biotechnology), then rotated for another 16 h at 4 °C. The lysates were incubated with G–Sepharose beads (17061801, GE Healthcare) for 2 h. Then, the beads were washed four times with lysis buffer. Bound proteins were eluted in sample buffer (100 mM Tris-HCl, pH 8.0) and analyzed by LC-MS/MS (see the section "On-beads digestion and preparation for proteome analysis by LC-MS/MS").

## Isolation of nuclei from the frontal cortex of mice

For proteome analysis, the frontal cortex of Xpo7^{+/+} and Xpo7^{+/−} mice at the ages of 1 month, 3 months, and 6 months, and Xpo7^{−/−} mice at the age of 1 month were dissociated using the gentleMACS Octo Dissociator instrument (Miltenyi) with Nuclei Extraction Buffer (130-128-024, Miltenyi) and Debris Removal Solution (130-109-398, Miltenyi) according to the manufacturer's protocol. The prepared nuclei pellets were filtered with 40 μm FLOWMI Tip Strainers (H13680-0040, Bel-Art Products).

## Sample preparation for proteome analysis by liquid chromatography–mass spectrometry (LC-MS/MS)

Isolated nuclei from the frontal cortex (1, 3, and 6 months of age from Xpo7^{+/+} and Xpo7^{+/−} mice, 1 month of age from Xpo7^{−/−} mice; 3 mice per group, prepared as above) and total frontal cortex samples (1, 3, and 6 months of age from Xpo7^{+/+} and Xpo7^{+/−} mice, 1 month from Xpo7^{−/−} mice; 3 mice per group) were further processed for analysis by LC-MS/MS as described in a previous study (Kawashima et al, 2022). In brief, cells were dissolved in 100 mM Tris-HCl (pH 8.0) containing 4% SDS and 20 mM NaCl using BIORUPTOR BR-II (SONIC BIO Co., Kanagawa, Japan). The Pierce™ BCA Protein Assay Kit (Thermo Fisher Scientific, WA, USA) at 500 ng/μL was quantified into 20 micrograms of extracted proteins. The protein extracts were reduced with 20 mM tris(2-carboxyethyl) phosphine for 10 min at 80 °C, followed by alkylation with 30 mM iodoacetamide for 30 min at room temperature in the dark. Protein purification and digestion were performed using the SP3 method (Hughes et al, 2019; Kawashima et al, 2022). Tryptic digestion was performed using 500 ng/μL Trypsin/Lys-C Mix (V5071, Promega, Madison, WI, USA) overnight at 37 °C. The digests were purified using GL-Tip SDB (7820-11200, GL Sciences, Tokyo, Japan) according to the manufacturer's protocol. The peptides were dissolved again in 2% ACN containing 0.1% TFA and quantified using the BCA assay at 150 ng/μL.

## On-beads digestion and preparation for proteome analysis by LC-MS/MS

On-beads digestion and preparation for proteome analysis by LC-MS/MS of immunoprecipitated samples were performed as previously described (Konno et al, 2024). In brief, 1.5 μL of 500 mM CaCl2 containing 2% lauryl maltose neopentyl glycol (Cat# NG310, Anatrace, Maumee, OH, USA) was added to the

beads soaked in 100 μL of 100 mM Tris-HCl (pH 8.0). Subsequently, 500 ng of trypsin (Trypsin Platinum, Cat# VA900, Promega, Madison, WI, USA) was gently mixed at 37 °C for 15 h to digest the proteins. The digested sample (supernatant) was collected in a new 1.5 mL tube and treated with 10 mM tris(2-carboxyethyl) phosphine (Thermo Fisher Scientific, Waltham, MA, USA) and 40 mM 2-chloroacetamide at 80 °C for 15 min. The alkylated sample, cooled to room temperature, was then acidified with 20 μL of 5% trifluoroacetic acid (TFA, Thermo Fisher Scientific) and desalted using GL-Tip SDB (Cat# 7820-11200, GL Sciences Inc., Tokyo, Japan) according to the manufacturer's protocol. The sample was dried in a centrifugal evaporator (miVac Duo concentrator, Genevac Ltd., Ipswich, UK) and redissolved in 0.1% TFA containing 0.01% decyl maltose neopentyl glycol (Cat# NG322, Anatrace).

## Liquid chromatography–mass spectrometry (LC-MS/MS)

For LC-MS/MS analysis, digested peptides from isolated nuclei from the frontal cortex and total frontal cortex samples were loaded directly onto a 75 μm × 12 cm nanoLC nano-capillary column (Nikkyo Technos Co., Ltd., Tokyo, Japan) at 40 °C and then separated with a 30-minute gradient (mobile phase A: 0.1% formic acid (FA) in water, mobile phase B: 0.1% FA in 80% acetonitrile (ACN)). The gradient was 0 min at 8% B, increasing to 70% B at 30 min, with a 200 nL/min flow rate using an UltiMate 3000 RSLCnano LC system (Thermo Fisher Scientific). The eluted peptides were detected using a quadrupole Orbitrap Exploris 480 hybrid mass spectrometer (Thermo Fisher Scientific) with a standard window data-independent acquisition (DIA). The MS1 scan range was set to a full scan with $m/z$ 490–745 at a mass resolution of 15,000, an Auto Gain Control (AGC) target of $3 \times 10^6$, and a maximum injection time of 23 ms. The MS2 scans were collected for ions greater than $m/z$ 200 at a resolution of 30,000, with an AGC target of $3 \times 10^6$, a maximum "auto" injection time, and a fixed normalized collision energy of 28%. The isolation width for MS2 was set to 4 Th, and for the 500–740 m/z window pattern, an optimized window arrangement was used in Scaffold DIA (Proteome Software, Inc., Portland, OR, USA). DIA-MS with a narrow isolation width of 4 Th compared with the conventional 10–20 Th enables more precise quantification of the identified proteins. High peak selectivity in the MS/MS analysis is considered superior to the western blot analysis for quantitative accuracy (Aebersold et al, 2013; Liebler and Zimmerman, 2013).

## LC-MS/MS analysis of immunoprecipitated samples

Peptides were directly injected into a 75 μm × 12 cm nanoLC nano-capillary column (Nikkyo Technos Co., Ltd., Tokyo, Japan) at 50 °C and separated using a 70-min gradient at a flow rate of 200 nL/min with an UltiMate 3000 RSLCnano LC system (Thermo Fisher Scientific). Peptides eluting from the column were analyzed using a Q Exactive HF-X (Thermo Fisher Scientific) for data-independent acquisition mass spectrometry (DIA-MS, also known as SWATH-MS). MS1 spectra were collected in the range of 495–745 $m/z$ at a resolution of 30,000, with an AGC target of 3e6 and a maximum injection time of 55 ms. We collected MS2 spectra for the ions above 200 $m/z$ at a resolution of 30,000, with an AGC target of 3e6, a maximum injection time set to "auto," and a normalized collision

energy of 23%. The isolation width for MS2 was set to 4 Th, and an optimized window arrangement for the 500–740 $m/z$ range was used in Scaffold DIA (Proteome Software, Inc., Portland, OR, USA).

MS files were searched against an in silico human spectral library using DIA-NN (version 1.8.1, https://github.com/vdemichev/DiaNN) (Demichev et al, 2020). DIA-NN generated a spectral library from the human protein sequence UniProt database (proteome ID UP000005640, 20,591 entries, downloaded on March 7, 2023). The parameters for generating the spectral library were as follows: digestion enzyme, trypsin; missed cleavages, 1; peptide length range, 7–45; precursor charge range, 2–4; precursor $m/z$ range, 490–750; and fragment ion m/z range, 200–1800. "FASTA digest for library-free search/library g, generation;" "deep learning-based spectra, RTs, and IMs prediction;" "N-terminal methionine excision;" and "C-terminal carbamidomethylation" was enabled. The DIA-NN search parameters were as follows: mass accuracy, 10 ppm; MS1 accuracy, 10 ppm; protein inference, genes; neural network classifiers, single-pass mode; quantification strategy, robust LC (high precision); and cross-run normalization, off. "Unrelated runs," "use isotopologues," "heuristic protein inference," and "no shared spectra" were enabled. The match between runs (MBR) was turned off. The protein identification threshold was 1% or less for precursor and protein false discovery rates (FDRs).

## Data processing of LC-MS/MS

The raw data were searched against an in silico predicted spectral library using DIA-NN (version 1.8.1, https://github.com/vdemichev/DiaNN) (Demichev et al, 2020). A spectral library was initially generated using DIA-NN from the human protein sequence database (UniProt ID UP000005640, reviewed, canonical, 20,381 entries). The search parameters for DIA-NN were as follows: protease, trypsin; missed cleavages, 1; peptide length range, 7–45; precursor charge range, 2–4; precursor mass range, 495–745 $m/z$; fragment ion $m/z$ range, 200–1800; mass accuracy, 10 ppm; static modification, cysteine carbamidomethylation. The following settings were enabled: "Heuristic protein inference," "Use isotopologues," "Match Between Runs (MBR)," and "No shared spectra." Additional commands included: "mass acc cal 10," "peak translation," and "matrix spec q." The protein identification threshold was set to less than 1% for both peptide and protein FDRs. Statistical calculations and Pearson correlation coefficient heatmap analysis with hierarchical clustering were performed using Perseus v1.6.15.0 (Tyanova et al, 2016).

## Single-nucleus RNA sequencing

The cell nuclei were collected from the sections containing both the frontal cortex and striatum (Bregma 1.30–3.30, excluding the olfactory area) of 12 mice [4 groups ($Xpo7^{+/+}$ and $Xpo7^{+/-}$ mice at 3 and 6 months of age); three mice/group]. A total of 27,241 nuclei were analyzed (Table EV1). The detailed methodologies are described below. Myelin and other debris were removed as described in the "Isolation of nuclei from the frontal cortex of mice" section. According to the manufacturer's protocol (Chromium Next GEM Single Cell Multiome ATAC + Gene Expression Reagent Kits User Guide (CG000375 • Rev C)), single nuclei and

barcoded beads were captured into droplets using the 10x Chromium platform. The Chromium Next GEM Single Cell Multiome ATAC + Gene Expression Reagent Bundle (10x Genomics, PN-1000285) was used for library preparation. After nucleus capture, reverse transcription, cDNA amplification, and sequencing library preparation were performed. The libraries were sequenced on an Illumina HiSeq 2500 sequencer with paired-end sequencing (Read1, 26 bp; Index, 8 bp; Read2, 98 bp). Mapping and annotation were performed using Cell Ranger Arc (version 2.0.2) with the mouse reference genome (mm10).

## Processing of scMultiome data

The Seurat package in R (version 5.0.1) (Butler et al, 2018) was used for subsequent analysis. For quality control, we removed cells with nFeature_RNA over 10,000 or less than 1000, nFeature_ATAC over 25,000 or less than 1000, and those with >5% mitochondrial counts. We filtered doublets using the cxds_bcds_hybrid function in the scds R package (Bais and Kosta, 2020) and removed ambient RNA using DecontX (Yang et al, 2020). The gene expression UMI count data were normalized using SCTransform (Hafemeister and Satija, 2019), regressing out mitochondrial counts, followed by dimensionality reduction with the RunPCA function in Seurat. We integrated the gene expression datasets across samples using SelectIntegrationFeatures ($n = 3000$), PrepSCTIntegration, FindIntegrationAnchors, and IntegrateData. For normalization and dimensionality reduction of the chromatin accessibility data, we used the Signac R package (version 1.12.0) (Stuart et al, 2021). The cell-peak matrix was binarized and normalized using the RunT-FIDF function. Singular value decomposition was performed using the RunSVD function on accessible peaks with a minimum cutoff of 10 (FindTopFeatures (min. cutoff = 10)). We found an anchor set between samples using FindIntegrationAnchors and integrated the chromatin accessibility data using the IntegrateEmbeddings function in Signac. RunUMAP was executed using the integrated embeddings. A joint neighbor graph was computed using Seurat's weighted nearest neighbor methods based on chromatin accessibility and gene expression data. Graph-based clustering was performed on PCA components 1–30 and LSI components 2–30. After RunUMAP, we calculated the supervised projection of the integrated object using RunSPCA with RNA gene expression data.

## Single-nucleus RNA sequencing data analysis

For cell-type identification, we used FindNeighbors and FindClusters with 30 PCA dimensions and a resolution of 2.0. Based on the expression levels of known cell-type markers, each cluster was ultimately assigned to six main cell types. Clusters expressing more than one cell-type marker and clusters with fewer than 100 cells were excluded. Excitatory and inhibitory neurons were clustered in the same manner as above and assigned to eight and eleven cell types, respectively, based on cell-type markers from the literature (Bhattacherjee et al, 2019; Clark et al, 2020; Zhang et al, 2021; Farsi et al, 2023; Yao et al, 2023). The Wilcoxon rank-sum test incorporated into the FindMarkers function from Seurat was used for the differential expression analysis. A gene was considered differentially expressed if the log fold change of the average expression between two groups was greater than 0.25 or less than −0.25, and the Bonferroni adjusted $p$-value was less than 0.05. We compared one cell type with the others to identify cell type-specific DEGs.

## Hypergeometric distribution

We conducted a hypergeometric distribution test to assess the overlap between SCHEMA molecules and DEGs identified from Single-nucleus RNA sequencing. The statistical significance $P$ and the expected value $E$ were calculated as follows:

$$P(X = x) = \frac{\binom{m}{x}\binom{N-m}{n-x}}{\binom{N}{n}} \tag{1}$$

$$E(X) = \frac{mn}{N} \tag{2}$$

Here, $x$ is the number of genes overlapping between SCHEMA molecules and DEGs, and $m$ and $n$ are the numbers of SCHEMA molecules and DEGs, respectively. $N$ is the total number of genes analyzed.

## Mouse protein interaction network analysis

We obtained protein interaction data for the mouse (Mus musculus, NCBI taxonomy Id: 10090) from the STRING database (Szklarczyk et al, 2023). Using this data, we built a protein interaction network (PIN) composed of Xpo7, Xpo7$^{+/-}$-direct-effect molecules, and their interacting partners. We only included protein interactions with a confidence score greater than 0.4. The constructed PINs were divided into modules using the Infomap algorithm (Rosvall and Bergstrom et al, 2008).

## Statistical analysis

single-nucleus RNA sequencing analysis, ATAC analysis, and hypergeometric distribution analysis were conducted using R software version 4.3.2. Other statistical analyses were performed using GraphPad Prism 8.4.3 (GraphPad Software, Inc, CA, USA). Data groups were compared using Tukey's Honestly Significant Difference (HSD) test unless specified in the figure legends. Sample sizes were determined based on previous studies (Shiwaku et al, 2023), and all experiments were randomized. A $P$ value < 0.05 was considered statistically significant. The exact values of $n$, the definitions of center dispersion, and precision measures are provided in the figure legends. Gene functional enrichment analysis was done using the Metascape database (http://metascape.org/) (Zhou et al, 2019).

# Data availability

The source data of single nucleus RNA sequencing presented in Figs. 4, 5, and 6 are available in the following URL: https://ddbj.nig.ac.jp/search/entry/bioproject/PRJDB19710.

The source data of this paper are collected in the following database record: biostudies:S-SCDT-10_1038-S44319-024-00362-9.

## Peer review information

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

## Acknowledgements

This work was supported by SENSHIN Medical Research Foundation, a Grant-in-Aid for Scientific Research from Japan Society for Promotion of Science (JSPS) (19K08011, 22K07553), Japan Agency for Medical Research and Development (AMED) (JP22wm0525036, JP24wm0625308, JP24wm0625312, JP24wm0625415), and FOREST (JPMJFR231U) to HS, KAKENHI JP (23H04979, 24K02379) from the Ministry of Education, Culture, Sports, Science and Technology of Japan and CREST (JPMJCR22P3) by from the Japan Science and Technology Agency to HT, AMED (JP23wm0525019) to MK, Multidisciplinary Frontier Brain and Neuroscience Discoveries (Brain/MINDS 2.0) (JP24wm0625308) from AMED to KFT. The authors thank Ms. Harumi Ishikubo and Ms. Takako Usami for contributing to generating XPO7$^{+/-}$ mice.

## Author contributions

**Saori Toyoda**: Data curation; Software; Formal analysis; Validation; Investigation; Methodology; Writing—original draft; Writing—review and editing. **Masataka Kikuchi**: Resources; Data curation; Software; Formal analysis; Funding acquisition; Validation; Investigation; Methodology; Writing—original draft; Writing—review and editing. **Yoshihumi Abe**: Data curation; Formal analysis; Validation; Investigation; Methodology; Writing—original draft; Writing—review and editing. **Kyosei Tashiro**: Data curation; Formal analysis; Validation; Investigation; Methodology. **Takehisa Handa**: Formal analysis; Investigation. **Shingo Katayama**: Formal analysis; Investigation. **Yukiko Motokawa**: Formal analysis; Investigation. **Kenji F Tanaka**: Supervision; Funding acquisition. **Hidehiko Takahashi**: Supervision; Funding acquisition. **Hiroki Shiwaku**: Conceptualization; Resources; Data curation; Software; Formal analysis; Supervision; Funding acquisition; Validation; Investigation; Visualization; Methodology; Writing—original draft; Project administration; Writing—review and editing.

Source data underlying figure panels in this paper may have individual authorship assigned. Where available, figure panel/source data authorship is listed in the following database record: biostudies:S-SCDT-10_1038-S44319-024-00362-9.

## Disclosure and competing interests statement

The authors declare no competing interests.

# Expanded View Figures

**Figure EV1.   Genotyping, Xpo7 expression analysis, and brain structure analysis of Xpo7 knockout mice.**

(**A**) The result of genotyping PCR of Xpo7 knockout mice. (**B**) The amino acid sequence of Xpo7 (exon 1–6). Six types of peptide fragments (red or blue) were detected from the sequence including 1–199 aa (exons 1–6) from $Xpo7^{+/+}$ and $Xpo7^{+/-}$ mice using LC-MS/MS, but not from $Xpo7^{-/-}$ mice. (**C**) H&E staining of coronal section at the bregma of the brain at 3 months of age. Bar: 1 mm. (**D**) Immunostaining for layer-specific markers, Cux1 and Tbr1, reveals normal cortical layer organization in $Xpo7^{+/-}$ mice at 3 months of age. Bar: 100 μm.

**A**

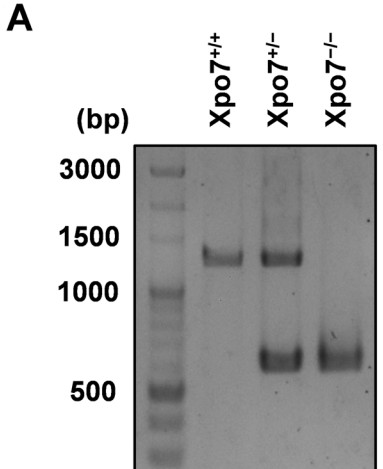

**B**

The amino acid sequence of Xpo7 (exon 1–6)

MADHVQSLA**QLENLCKQLYETTDTTTR**LQAEKALVEFTNSPDCLSKCQLL
LER**GSSSYSQLLAATCLTK**LVSRTNNPLPLEQRIDIRNYVLNYLATRPK**L
ATFVTQALIQLYAR**ITK**LGWFDCQK**DDYVFRNAITDVTRFLQDSVEYCII
GVTILSQLTNEINQADTTHPLTKHRKIASSFR**DSSLFDIFTLSCNLLK**Q

**C**

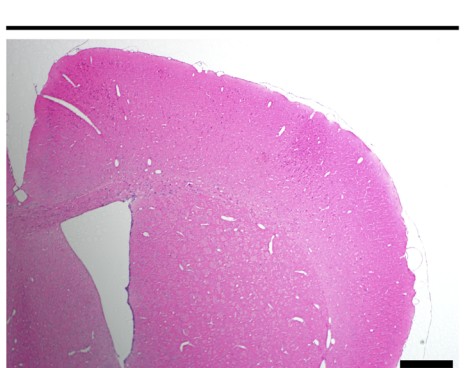

Xpo7<sup>+/+</sup>                      Xpo7<sup>+/−</sup>

**D**

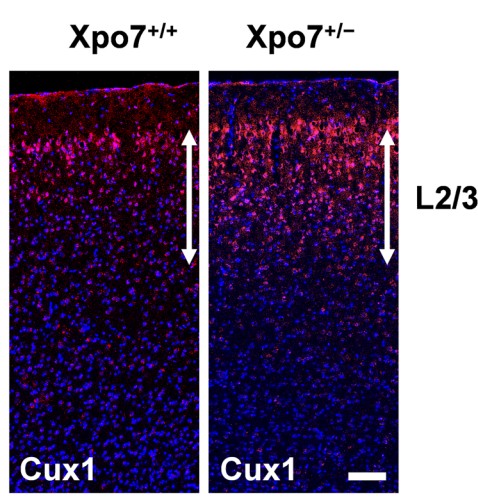

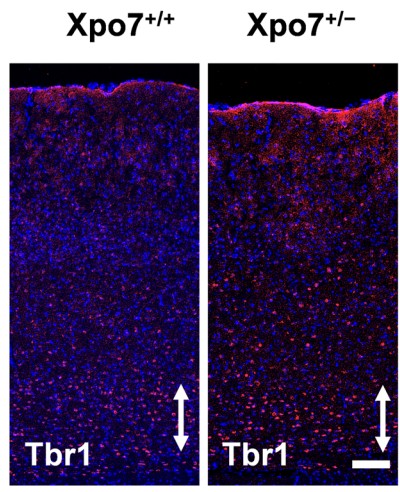

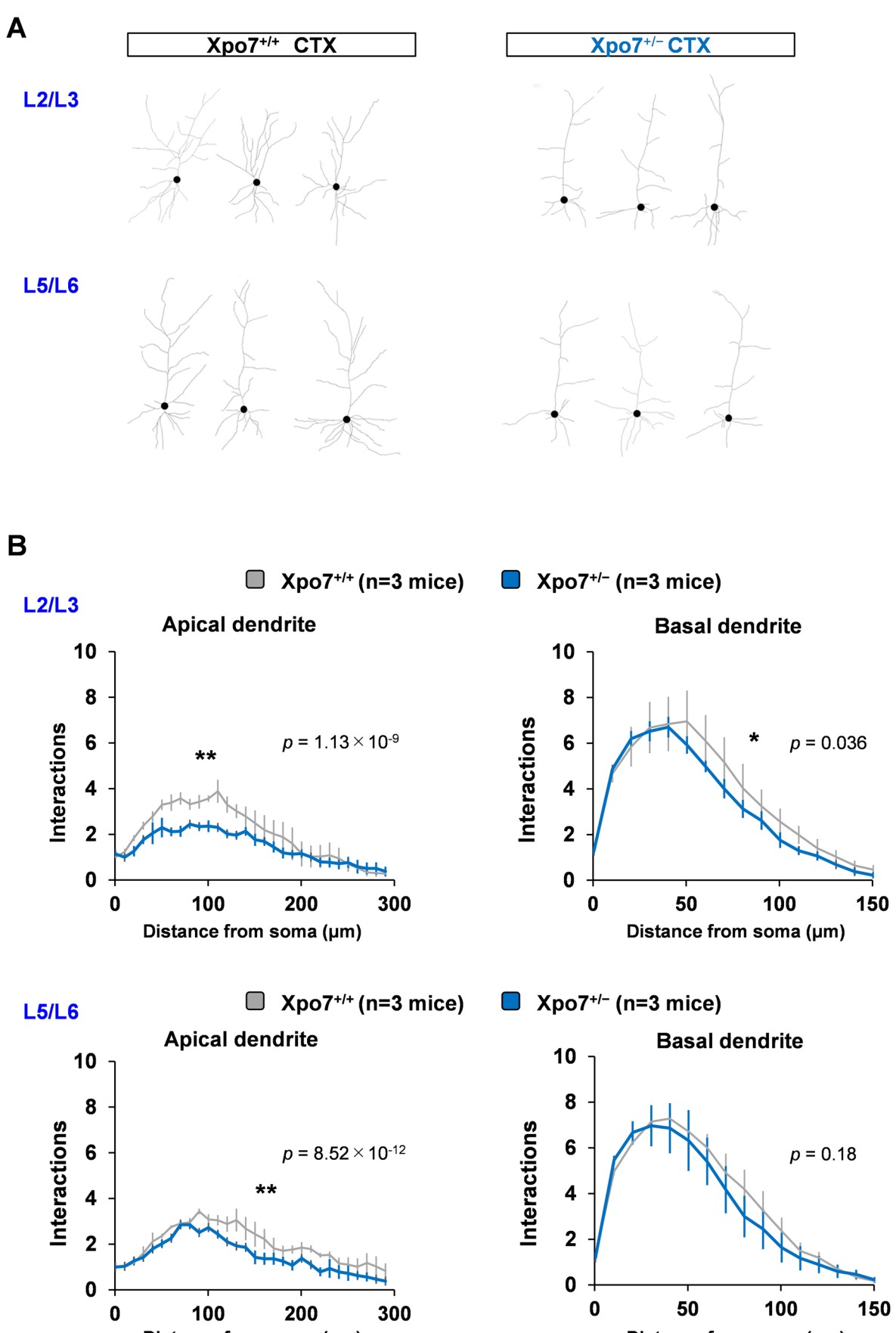

**Figure EV2. Dendrite analyses from the Golgi staining data revealed fewer dendrite branches in Xpo7$^{+/-}$ mice.**

(A) Representative neurons from L2/L3 and L5/L6 of the frontal cortex at three months of age. (B) Sholl analyses of apical and distal dendrites of L2/L3 neurons and L5/L6 neurons of the frontal cortex at three months of age. *$P < 0.05$, **$P < 0.01$ ($N = 3$ mice. For each animal, 6–12 pyramidal neurons from either layers 2–3 or layers 5–6 were analyzed. Two-way repeated ANOVA test). Data are expressed as the mean ± s.e.m.

**A**

L2/L3

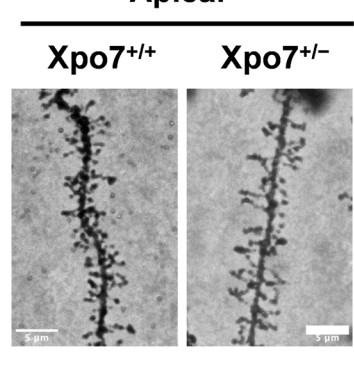
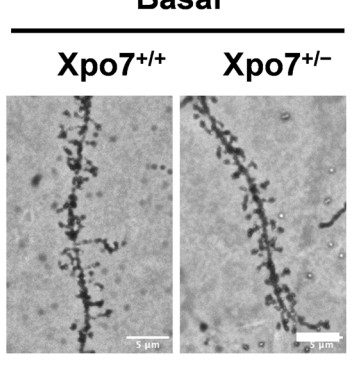

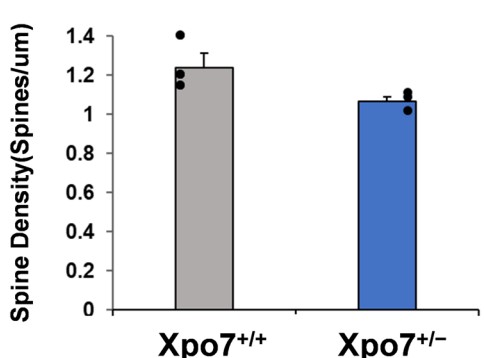
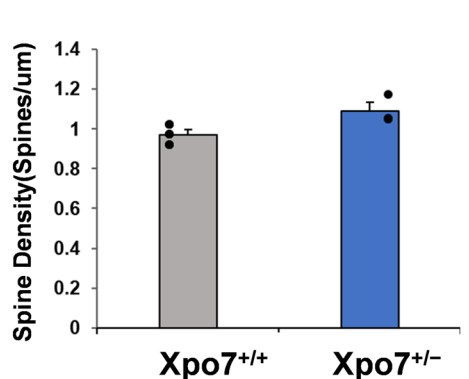

**B**

L5/L6

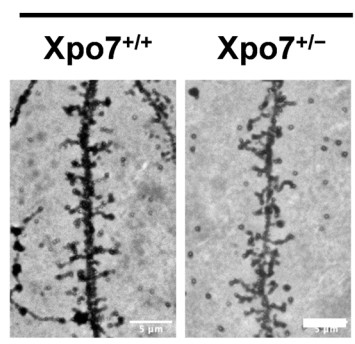
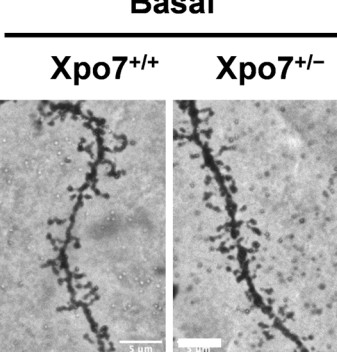

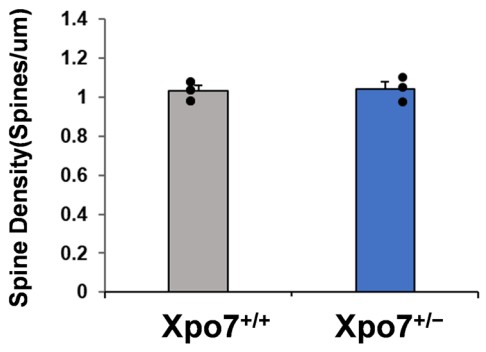
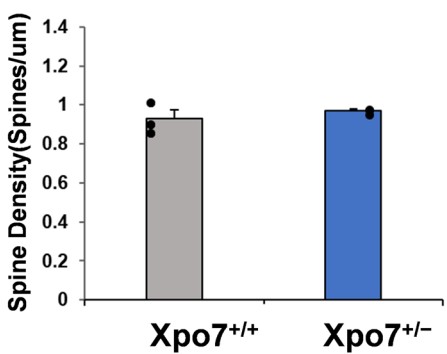

◀ **Figure EV3. Spine analyses from Golgi staining data revealed no significant change in spine densities per dendrite in Xpo7$^{+/-}$ mice.**

(A) Representative spines from L2/L3 neurons of frontal cortex at three months of age. The graphs show quantitative analysis of spine number. ($N = 3$ mice, 6–14 neurons/mice) Bar: 5 μm. Data are expressed as the mean ± s.e.m. (B) Representative spines from L5/L6 neurons of frontal cortex at three months of age. The graphs show quantitative analysis of spine number. ($N = 3$ mice, 6–14 neurons/mice) Bar: 5 μm. Data are expressed as the mean ± s.e.m.

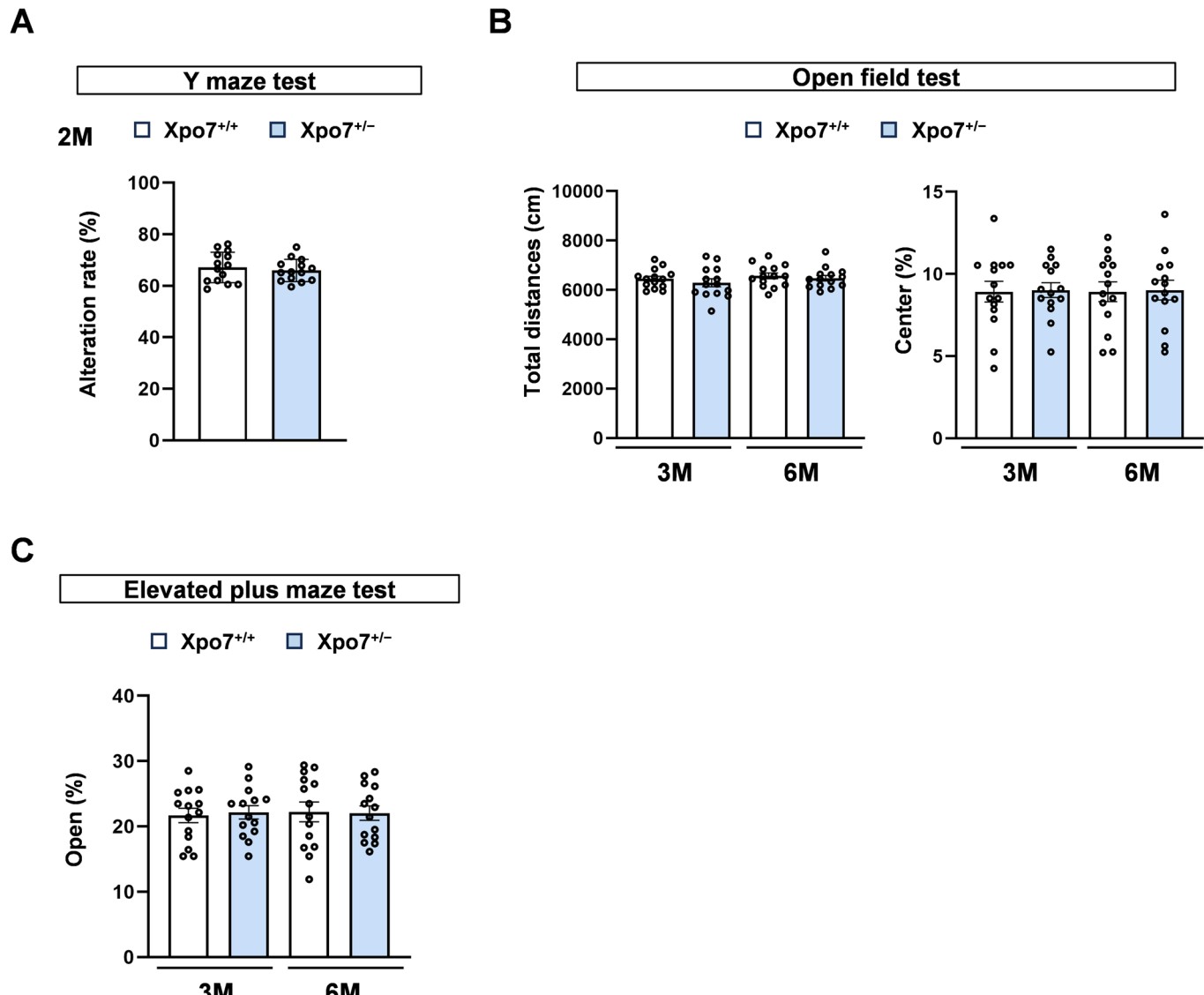

**Figure EV4. Behavioral analysis of XPO7$^{+/-}$ mice.**

(A) Alteration ratios in the Y maze test with Xpo7$^{+/+}$ and Xpo7$^{+/-}$ mice at 2 months of age. There was no significant difference between the groups. ($N = 14$ for each group; Tukey's HSD test). Data are expressed as the mean ± s.e.m. (B) Total distances (left) and Time in center region (right) in the open-field test. There was no significant difference between the groups. ($N = 14$ for each group; Tukey's HSD test). Data are expressed as the mean ± s.e.m. (C) Time in the open arm regions in the elevated plus maze test. There was no significant difference between the groups. ($N = 14$ for each group; Tukey's HSD test). Data are expressed as the mean ± s.e.m.

# A

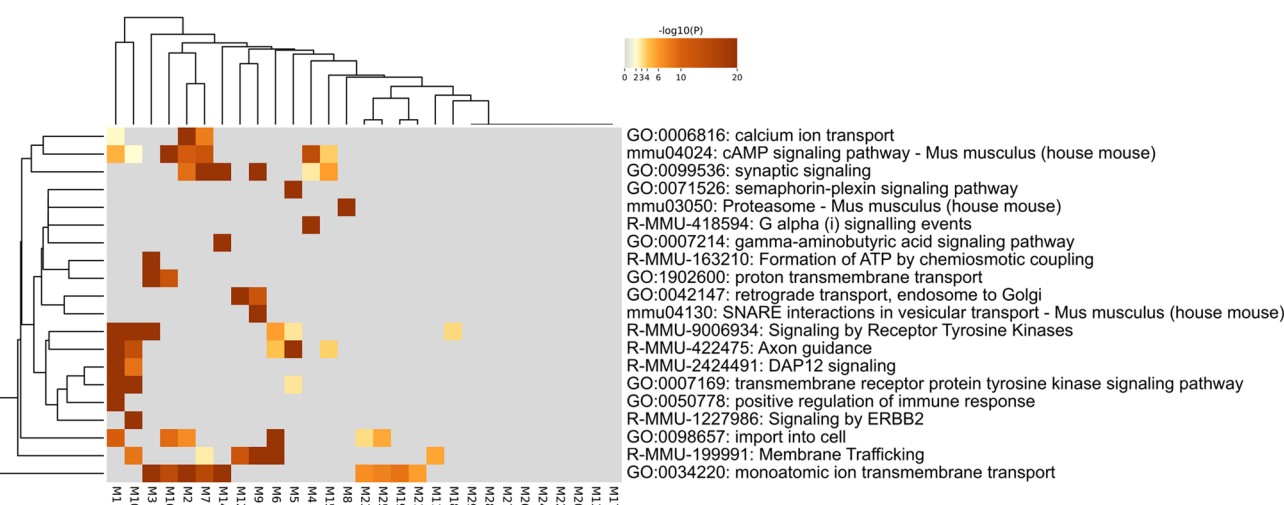

**Figure EV5.  Protein interaction network analysis 45 Xpo7 direct-effect molecules.**

(**A**) The 1-hop protein–protein interaction analysis with Xpo7 revealed 847 molecules. the Infomap algorithm revealed 28 modules (M1 – M28). (**B**) Gene Ontology (GO) analysis of (**A**). These analyses revealed that 45 XPO7 direct-effect molecules are associated with transmembrane transport (M10, M7, M12, M9, M6), immune response (M1), axon guidance (M1, M10, M6, M5, M15), synaptic signaling (M2, M7, M14, M9, M4, M15) and proteasome (M8).

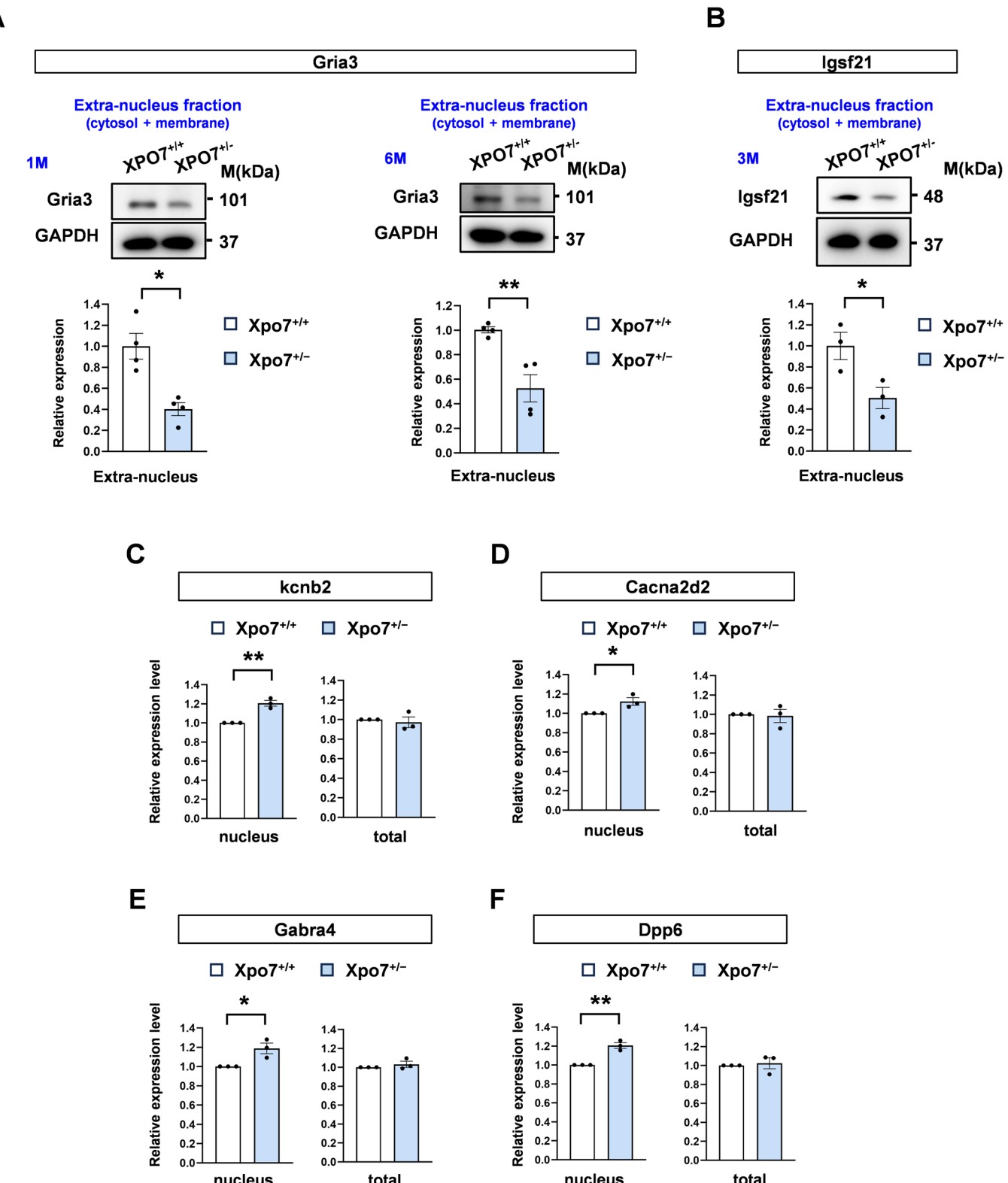

◀ **Figure EV6. Detailed analysis of Xpo7-binding and Xpo7 haploinsufficiency-affected molecules.**

(A) Western blot analysis of Gria3 in the extranuclear fraction from the frontal cortex of Xpo7$^{+/+}$ and Xpo7$^{+/-}$ mice at 1 and 6 months of age. **$P < 0.01$ (1 M $p = 0.0049$; 6 M $P = 0.0057$; $N = 4$ experiments; Tukey's HSD test). Data are expressed as the mean ± s.e.m. (B) Western blot analysis of Gria3 in the extranuclear fraction from the frontal cortex of Xpo7$^{+/+}$ and Xpo7$^{+/-}$ mice at 3 months of age. *$P < 0.05$ ($P = 0.039$; $N = 3$ experiments; Tukey's HSD test). Data are expressed as the mean ± s.e.m. (C–F) Relative expression of kcnb2, Cacna2d2, Gabra4, and DPP6 in nuclei and whole cells from the frontal cortex of Xpo7$^{+/+}$ and Xpo7$^{+/-}$ mice by LC-MS/MS analysis. *$P < 0.05$, **$P < 0.01$ (kcnb2 $P = 0.0024$; Cacna2d2 $P = 0.035$; Gabra4 $P = 0.024$; Dpp6 $P = 0.0024$; $N = 3$ experiments, 3 mice per group/experiment; Tukey's HSD test). Data are expressed as the mean ± s.e.m. As described in Fig. 3, the LC-MS/MS analysis data were derived from pairs of mice at 1, 3, and 6 months of age.

## A

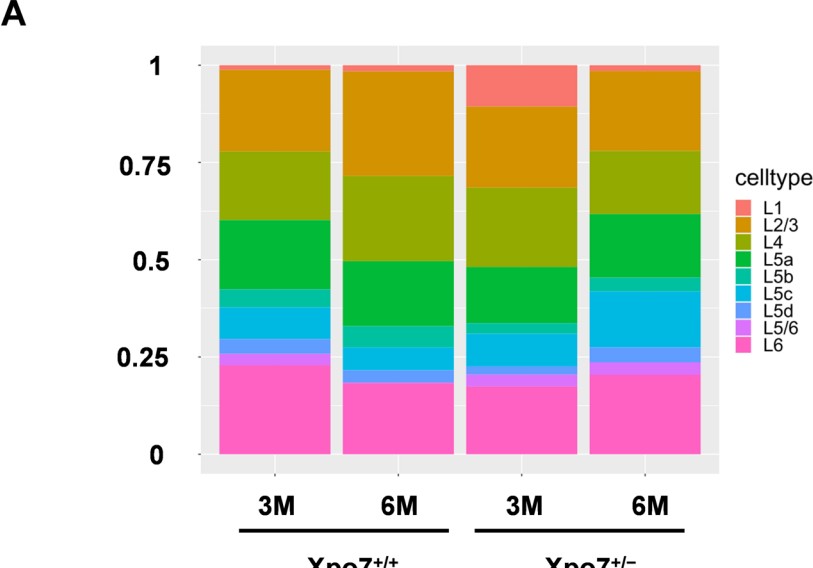

## B

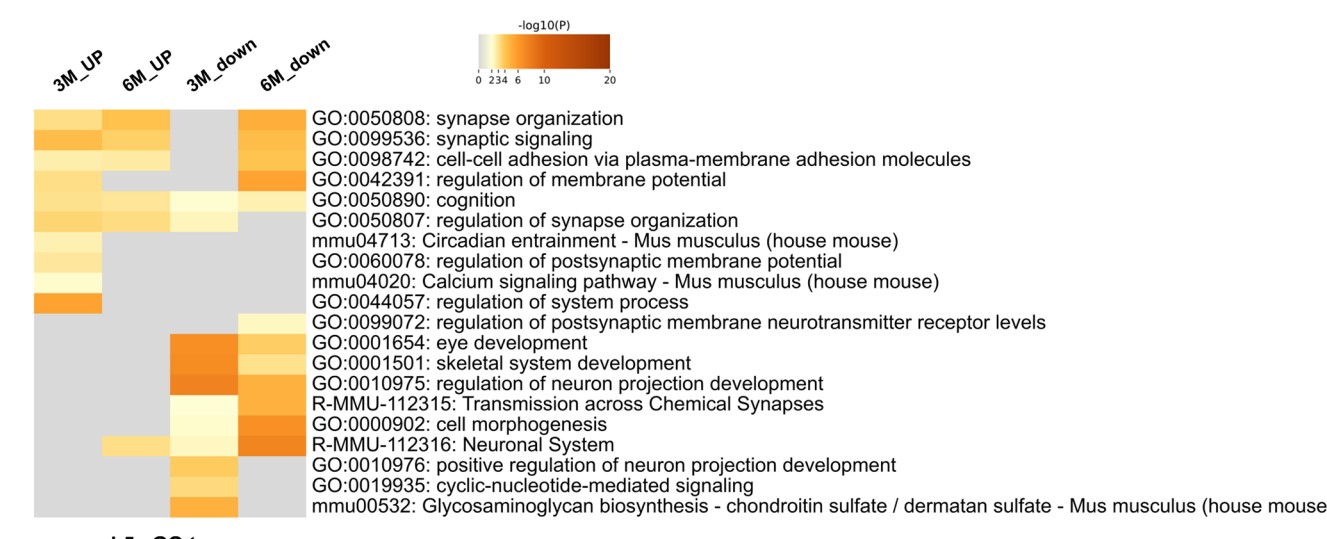

## C

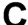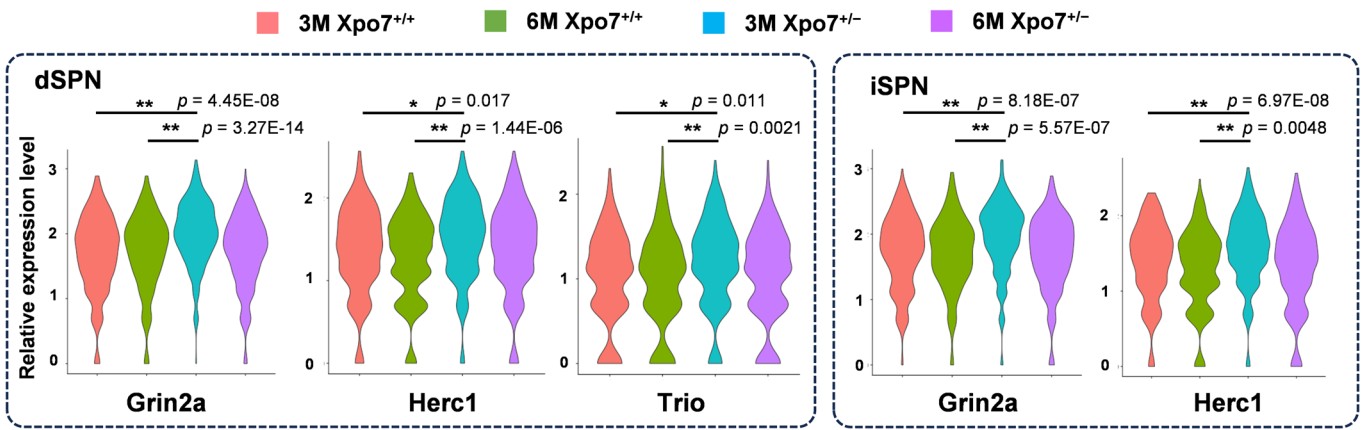

◄ **Figure EV7. Detailed analysis for single-nucleus RNA sequencing analysis.**

(A) The ratio of neurons in the frontal cortical layers by single-nucleus RNA sequencing analysis. (B) GO analysis of DEGs of L5c excitatory neurons. To evaluate the functional enrichment of a given gene list, the metascape performed an accumulative hypergeometric test or Fisher's exact test to calculate $P$ values and enrichment factors for each ontology category. (C) Relative mRNA expression levels in dSPN and iSPN. Relative mRNA expression levels of Grin2A, Herc1, and TRIO **$P$ < 0.01 (dSPN 3 M Xpo7$^{+/+}$ $N$ = 540 cells, 6 M Xpo7$^{+/+}$ $N$ = 544 cells, 3 M Xpo7$^{+/-}$ $N$ = 481 cells, 6 M Xpo7$^{+/-}$ $N$ = 319 cells; iSPN 3 M Xpo7$^{+/+}$ $N$ = 442 cells, 6 M Xpo7$^{+/+}$ $N$ = 487 cells, 3 M Xpo7$^{+/-}$ $N$ = 464 cells, 6 M Xpo7$^{+/-}$ $N$ = 250 cells; Wilcoxon rank-sum test).

