## [Peer Review File · EMBO Reports]

Schizophrenia-related Xpo7 haploinsufficiency leads to behavioral and nuclear transport pathologies

Saori Toyoda, Masataka Kikuchi, Yoshihumi Abe, Kyosei Tashiro, Takehisa Handa, Shingo Katayama, Yukiko Motokawa, Kenji Tanaka, Hidehiko Takahashi, and Hiroki Shiwaku

Corresponding author: Hiroki Shiwaku (shiwaku.npat@mri.tmd.ac.jp)

Review Timeline:

Submission Date:	23rd Jun 24
Editorial Decision:	26th Jul 24
Revision Received:	9th Nov 24
Editorial Decision:	3rd Dec 24
Revision Received:	9th Dec 24
Accepted:	18th Dec 24

Editor: Esther Schnapp

Transaction Report:

Dear Prof. Shiwaku,

Thank you for the submission of your manuscript to EMBO reports. I could only secure 2 referees for it and we have received their reports now (included below).

As you will see, the referees acknowledge that the findings are potentially interesting. However, they also have several suggestions for how the study could be improved, and I think all are good and should be addressed, except where referees indicate that the revision is optional. It would be helpful to know what you think and to discuss the exact revision requirements, also in a video chat, if you wish. I would like to suggest that you send me a proposed revision plan where you outline what you can and cannot address, and we can take it from there. Please let me know what you think.

I would thus like to invite you to revise your manuscript with the understanding that the referee concerns must be fully addressed and their suggestions taken on board. Please address all referee concerns in a complete point-by-point response. Acceptance of the manuscript will depend on a positive outcome of a second round of review. It is EMBO reports policy to allow a single round of major revision only and acceptance or rejection of the manuscript will therefore depend on the completeness of your responses included in the next, final version of the manuscript.

We realize that it is difficult to revise to a specific deadline. In the interest of protecting the conceptual advance provided by the work, we recommend a revision within 3 months (26th Oct 2024). Please discuss the revision progress ahead of this time with the editor if you require more time to complete the revisions.

- 1) A data availability section providing access to data deposited in public databases is missing. If you have not deposited any data, please add a sentence to the data availability section that explains that.
- 2) Your manuscript contains statistics and error bars based on $n=2$. Please use scatter blots in these cases. No statistics should be calculated if $n=2$.

3) We replaced Supplementary Information with Expanded View (EV) Figures and Tables that are collapsible/expandable online. A maximum of 5 EV Figures can be typeset. EV Figures should be cited as 'Figure EV1, Figure EV2' etc... in the text and their respective legends should be included in the main text after the legends of regular figures.

5) a complete author checklist, which you can download from our author guidelines <https://www.embopress.org/page/journal/14693178/authorguide>. Please insert information in the checklist that is also reflected in the manuscript. The completed author checklist will also be part of the RPF.

6) Please note that all corresponding authors are required to supply an ORCID ID for their name upon submission of a revised manuscript (<https://orcid.org/>). Please find instructions on how to link your ORCID ID to your account in our manuscript tracking system in our Author guidelines <https://www.embopress.org/page/journal/14693178/authorguide#authorshipguidelines>

12) All Materials and Methods need to be described in the main text using our 'Structured Methods' format, which is required for all research articles. According to this format, the Methods section includes a Reagents and Tools Table (listing key reagents, experimental models, software and relevant equipment and including their sources and relevant identifiers) followed by a Methods and Protocols section describing the methods using a step-by-step protocol format. The aim is to facilitate adoption of the methodologies across labs. More information on how to adhere to this format as well as a downloadable template (.docx) for the Reagents and Tools Table can be found in our author guidelines:
<https://www.embopress.org/page/journal/14693178/authorguide#structuredmethods>.

An example of a Method paper with Structured Methods can be found here: <https://www.embopress.org/doi/full/10.1038/s44320-024-00037-6#sec-4>

I look forward to seeing a revised form of your manuscript when it is ready.

Yours sincerely,

Referee #1:

The study conducted by Toyoda et al. employs mice mutant for the exportin 7 (XPO7) gene, one of the top 10 genes identified by the Schizophrenia Exome Sequencing Meta-Analysis (SCHEMA) consortium as carrying protein-truncating variants that increase the risk of schizophrenia. The authors report that mutant mice display cognitive deficits, they identify XPO7 interacting proteins that are affected in mutant mice as well as genes whose expression is altered in various cell types in the brains of these mice. While this work is of potential interest to the field, there are several issues that should be addressed to improve the manuscript.

1. In some parts of the manuscript, figure legends do not fit with the presented data (cp. EV1C-D) nor the indicated animal number reflects the single data points shown in the graphs (cp. Fig.1C-F). Since the methods section contains limited information about animal cohorts used, this discrepancy creates confusion. The authors do not provide details about the sex of the animals used in this study about the details of the behavioral assays (e.g. how long the break between Trial 1 and 2 was when investigating memory deficits). Also, the y-axis in some graphs displays consistent typos that should be carefully revised by the authors (e.g. "Reraltive exporession" "alterations").

2. The term schizophrenia-related behaviors is inaccurate. None of the behavioral/cognitive domains probed by the authors are specifically related to schizophrenia. They are merely used as proxies to probe the effect of the mutation on cognitive function which in general terms is affected in patients with schizophrenia

3. Among the affected behaviors, some are affected at 3mo but not at 2mo while one is presumably affected at 6mo. I am concerned that some of these age-dependent differences are due to uneven sample sizes. For example, the low number of animals used at 2mo (N=7) compared to the number used at 3mo and 6mo (N=20-30) in the Y-Maze assay raises the question if the 2mo assay is simply underpowered to detect significant changes. Similarly, the animal groups at 3mo demonstrate higher variation in the Social novelty preference (cp. Fig.1E) and, therefore, it cannot be ruled out that the social memory deficit is already present at 3mo and the observed trend could become significant by increasing the number of animals. In general, the authors need to be cautious about making claims about disease progression based on their behavioral data.

4. In a related issue, the authors attempt to rationalize their behavioral data (as well as accompanying gene expression data) in terms of disease progression from "onset (adolescent) to chronic phases (older age)". The rationale is confusing and needs to be clarified since 3mo in mouse lifetime is hardly equivalent to human adolescence. Also, despite what the authors state, unlike the age-dependent deterioration observed in degenerative disease, the clinical picture of schizophrenia is more or less stable over the years following onset. Of course, it is possible that, unlike idiopathic schizophrenia, patients with schizophrenia and XPO7 mutations display an atypical trajectory in their cognitive dysfunction that includes age-dependent deterioration. However, the authors need to be more careful with their statements, cite relevant literature and clarify their rationale behind analyzing disease progression.

5. To identify molecules with abnormal nuclear import or export in the frontal cortex of mutant mice, the authors use IP assays followed by LC-MS/MS analysis to isolate XPO7-interacting proteins whose abundance is altered in nuclei isolated from the frontal cortex of mutant mice. Use of homozygous mutant mice is commendable as it leads to a well-controlled experiment. However, the authors analyze their IP data using two analytical methods and two seemingly arbitrary cutoffs to identify 43 and 412 proteins respectively. The rationale behind these two analytical approaches and why there is an order of magnitude difference in the number of identified protein targets is not clearly stated. It is also unclear why they have to present both analytical methods.

6. Combination of IP and LC-MS/MS analysis identified 45 molecules interacting with XPO7. Only a small fraction of them (only two, CutC and Gria30, if I understand it correctly) were followed up by WB assays. Confirming a larger fraction will provide more confidence in the list of identified targets.

7. The age of the mice used in these analyses needs additional clarity. The authors state that mouse cerebral frontal cortex from XPO7^{+/+} and XPO7^{-/-} mice at the age of 1 month were used for the IP assay. For the LC-MS/MS assay nuclei were isolated from the frontal cortex of 1-, 3-, and 6-months old mice. It is unclear how data from various age mice were used in individual analyses and the comparison between IP and LC-MS/MS data.

8. Identification of Gria3, an AMPA receptor subunit and one of the top 10 SCHEMA genes, as a target of XPO7 haploinsufficiency is very interesting and potentially one of the most important findings of this study as it could nicely demonstrate interaction between two top SCHEMA genes. The authors observed elevated levels of Gria3 within nuclei in the frontal cortex of mutant mice via LC-MS/MS analysis, while the extranuclear Gria3 levels were decreased at 6 months. The authors need to explain more clearly what the increase in nuclear levels of Gria3 (as opposed to decrease) imply. Also why only 6-month-old mice were analyzed? It is important to know if such changes appear earlier as well. Although I understand this may be outside the scope of this work, I think the value of the paper will increase immensely if the authors could probe the function of AMPA receptors in these mice using electrophysiological assays.

9. There is no information on gross brain morphology and overall layer-specific density of neurons in adult mutant mice. Also there is no information on the impact of the XPO7 mutation on dendrites, and spines and axons of cortical neurons. Such data from morphological analysis will help interpret the single-nucleus RNA sequencing data. Along these lines the authors could use single-nucleus RNA sequencing data to determine and compare cell densities between wt and mutant mice.

10. Statements such as "we detected 284 gene expression changes that were linked to the behavioral phenotypes of XPO7" or "identified 15 genes in the frontal cortex related to disease progression" are inaccurate and require rephrasing. These are simply correlations as there is no experimental evidence linking these DEGs to either behavior or disease progression.

11. What is the rationale behind seeking overlaps between DEGs and XPO7 direct-effect molecules? If anything, such overlap confounds interpretation of the direct effect analysis.

Other issues:

It is not clear why the authors normalized all WT values to 1 in the relative expression data sets.

The infomap of the PPI network analysis shown in Fig.EV2A is too small and it is impossible to read the single network targets.

It is not really clear what role, if any, CutC and potential copper-induced damage can have in the development and progression of schizophrenia as discussed in the result section. Is there some relevant evidence in the literature?

Why does the Ethics section refer to human subjects?

Referee #2:

Toyoda et al. describe the generation of exportin 7, encoded by Xpo7, haploinsufficient and KO mice. XPO7 is a nuclear transport protein, identified as one of 10 SCHEMA (Schizophrenia Exome Sequencing Meta-Analysis) genes, carrying significant risk for schizophrenia. Two SCHEMA genes have been previously evaluated in mouse models, namely SETD1A and GRIN2A. This report will be the third SCHEMA gene evaluated in a mouse model. Here, Xpo7 haploinsufficient mice undergo behavioral and genomic analyses as a schizophrenia model. The authors find modest cognitive behavioral differences, which progress from adolescence to maturity. The genomics are the strength of the paper. The authors use intersectional approaches in single cell analyses to identify transported proteins that are differentially found in the nucleus, in the Xpo7 haploinsufficient mice. They find differentially regulated genes, which include other SCHEMA genes. Looking at the intersection of Xpo7 and progression-associated genes is particularly powerful, revealing an overlap with 4 of 10 SCHEMA genes. The authors perform cell-type and regional (cortex and striatum) analyses to localize gene expression changes. While no one mutant mouse can model schizophrenia fully, this study instantiates several associations that support the pathogenic schizophrenia risk of XPO7 deficiency.

Comments

1. Description of the generation of the XPO7 mutant mice is cursory. A gene map for their generation would be expected as Fig 1A. Description of the mice indicates that exon 7 is deleted. Is a truncated protein expressed from exons 1-6? Could this confer a toxic gain of function?

2. It should be discussed that Xpo7 is a nuclear exportin, and that the majority of its activity is nuclear export, not import.

3. Methods are lacking for assessing gene expression (allelic abundance), shown in Fig 1B, as well as genotyping.

4. There is no description of the dissections, what part of cortex was taken, no details on how cortical layers were separated, nor criteria used for isolating the striatum.
5. Nomenclature wise, in mice XPO7 should be written Xpo7 (human genes are all caps, mouse genes only first letter is capitalized).
6. Page 10, sentence, "The 45 molecules are presented in Fig. 3F. Gene Ontology (GO) analysis of these 45 molecules revealed that they contain many receptors and transporters involved in synapses (Fig. 3G)." Figure citations should be to Fig. 2, not 3.
7. Page 11, sentence, "We observed elevated levels of Gria3 within the frontal cortex of the nucleus of XPO7+/? mice through LC-MS/MS analysis (Fig. 3A)." Would be better, ...within the nuclei of frontal cortical cells in Xpo7+./- mice through...
8. Page 11, sentence, "CutC. Additionally, Fyn (Odds ratio 10.7, p = 0.000596) and Slitrk5 (Odds ratio 6.69, p = 0.000917) were molecules linked to an increased susceptibility to schizophrenia (Singh et al., 2022)." It should be stated that these genes were upregulated.
9. Page 12, sentence, "Additionally, striatal pathology has been frequently observed alongside dopamine and cortical pathology in schizophrenia." It is not clear what alongside means, nor is this statement referenced.
10. Page 13, sentence, "This is crucial as XPO7+/+ mice exhibit normal cognitive function and sociability at 6 months of age." This seems obvious, as XPO7+/+ mice are wild type mice.
11. Page 21, sentence, "The best approach for treating XPO7 haploinsufficiency would be to restore the expression of XPO7 itself." Yes, if the restoration were done in early embryogenesis; there is no assurance that restoring gene function in maturity will reverse developmental pathogenic effects (i.e. shutting the barn door after the horse has bolted).
12. Westerns are presented cutup, with the dynamic range differentially adjusted, based on background shade. They should be shown whole.
13. Fig 3A - should specify from 3 month old mice.
14. Fig EV3 uses the same color scale for up and down regulation, and does not include a fold scale look-up table.

Referee #1:**[Reviewer's comment 1]**

In some parts of the manuscript, figure legends do not fit with the presented data (cp. EV1C-D) nor the indicated animal number reflects the single data points shown in the graphs (cp. Fig.1C-F). Since the methods section contains limited information about animal cohorts used, this discrepancy creates confusion. The authors do not provide details about the sex of the animals used in this study about the details of the behavioral assays (e.g. how long the break between Trial 1 and 2 was when investigating memory deficits). Also, the y-axis in some graphs displays consistent typos that should be carefully revised by the authors (e.g. "Reraltive expoesion" "alterations").

[Answer]

Thank you very much for your comments. We corrected the errors in the figure legend with respect to the number of mice analyzed. We also corrected the errors in the graphs. We added a detailed description of the behavioral assays pointed out by the reviewer as follows.

All behavioral tests were performed using male C57BL/6J mice. (Line 466)

Novel object recognition test: The training session used two identical objects, and the mice were allowed a 10-minute exploration period. Immediately following this session, one of the objects was replaced with a novel object, and the mice were allowed to examine it again until they reached a total exploration time of 30 seconds. Omitting a break between sessions in the novel object recognition test aims to maintain a consistency in test structure with the three-chamber test. (Line 504-509)

[Reviewer's comment 2]

The term schizophrenia-related behaviors is inaccurate. None of the behavioral/cognitive domains probed by the authors are specifically related to schizophrenia. They are merely used as proxies to probe the effect of the mutation on cognitive function which in general terms is affected in patients with schizophrenia

[Answer]

Thank you very much for your valuable comments. We deleted the phrase "schizophrenia-related behaviors." We have changed the phrase to "cognitive and social behavioral

impairments” or just “behavioral pathologies.”

[Reviewer’s comment 3]

Among the affected behaviors, some are affected at 3mo but not at 2mo while one is presumably affected at 6mo. I am concerned that some of these age-dependent differences are due to uneven sample sizes. For example, the low number of animals used at 2mo (N=7) compared to the number used at 3mo and 6mo (N=20-30) in the Y-Maze assay raises the question if the 2mo assay is simply underpowered to detect significant changes. Similarly, the animal groups at 3mo demonstrate higher variation in the Social novelty preference (cp. Fig.1E) and, therefore, it cannot be ruled out that the social memory deficit is already present at 3mo and the observed trend could become significant by increasing the number of animals. In general, the authors need to be cautious about making claims about disease progression based on their behavioral data.

[Answer]

Thank you very much for your valuable comments. First, we increased the number of mice analyzed at two months of age (Fig. EV5A). As a result, we confirmed that there was no decrease in cognitive function at this time. We also increased the number of three-month-old mice for the three-chamber test, but there was no significant difference compared with the wild-type. As the reviewer indicated, however, we cannot completely rule out the possibility that there is a difference in abnormal behavior at three months; thus, we mentioned that there is a slight tendency toward abnormal behavior even at three months in the Result section. We also stated that it is necessary to conduct a wider range of behavioral tests in the future to confirm whether progressive abnormal behaviors exist in the Discussion section. We also reduced the emphasis on the phrase regarding the progressive behavioral abnormalities in mice.

In the three-chamber test, impaired social novelty preference was significantly observed in $Xpo7^{+/-}$ mice at 6 months of age, whereas it was not obvious at 3 months of age, indicating a possible progressive change in phenotype (Fig. 1E). There was a slight tendency toward impaired social novelty preference at 3 months of age. (Line 124-128)

It is necessary to conduct a wider range of behavioral tests in the future to confirm whether further progressive abnormal behaviors exist in $Xpo7^{+/-}$ mice. (Line 364-365)

[Reviewer's comment 4]

In a related issue, the authors attempt to rationalize their behavioral data (as well as accompanying gene expression data) in terms of disease progression from "onset (adolescent) to chronic phases (older age)". The rationale is confusing and needs to be clarified since 3mo in mouse lifetime is hardly equivalent to human adolescence. Also, despite what the authors state, unlike the age-dependent deterioration observed in degenerative disease, the clinical picture of schizophrenia is more or less stable over the years following onset. Of course, it is possible that, unlike idiopathic schizophrenia, patients with schizophrenia and XPO7 mutations display an atypical trajectory in their cognitive dysfunction that includes age-dependent deterioration. However, the authors need to be more careful with their statements, cite relevant literature and clarify their rationale behind analyzing disease progression.

[Answer]

Thank you very much for your important comment. First, we deleted the word "adolescent." Second, as the reviewer points out, large-scale analyses of patients with schizophrenia revealed that cognitive impairment is apparent at the onset, but this impairment remains stable throughout the course of the disease. However, the majority of these patients received treatment and a stable course may be the result of treatment. In fact, studies report that if the duration of the untreated period (DUP) is long, patients become refractory to treatment, suggesting the presence of progressive pathogenesis in schizophrenia (some molecular or cellular change may underly the changes in treatment response). It has also been reported that there is a subtype of patients with progressive cognitive decline over 10 years in the longitudinal study. Patients with poor or declining cognition were less likely to participate in follow-up assessments, suggesting that some patients with worse cognitive decline during the course exist. Relapse in schizophrenia is also associated with disease progression in terms of refractory treatment and a general decrease in brain volume. There is clinical evidence for the importance of early intervention and continuing treatment. Collectively, if treatment intervention is inappropriate, some patients may be refractory and we hypothesized that "progressive molecular pathogenesis" exists in such patients with schizophrenia.

Therefore, we changed the phrase "progressive symptoms" to the possible existence of "progressive molecular pathogenesis" for cases in which appropriate treatment is not being administered, and described the rationale for studying different stages of mice (3M and 6M) in the Introduction and Discussion sections as follows. As the reviewer also

pointed out, there are no reports on the course of patients with XPO7 variants. This point is important, so we also included it in the Discussion as follows.

Studies report that if the duration of the untreated period (DUP) is long, patients become refractory to treatment, suggesting the existence of progressive pathogenesis in schizophrenia (Marshall et al, 2005; Perkins et al, 2005). It has also been reported that there is a subtype of patients showing progressive cognitive decline over 10 years in longitudinal studies (Fett et al, 2020; Starzer et al, 2024; Zanelli et al, 2019). Those with poor or declining cognition were less likely to participate in follow-up assessments, suggesting that some patients with worse cognitive decline during the course exist (Starzer et al., 2024). Relapse in schizophrenia is also associated with disease progression in terms of treatment resistance and a general decrease in brain volume (Andreasen et al, 2013; Emsley et al, 2013). Thus, we hypothesize the existence of a progressive molecular pathogenesis. (Line 64-74)

Moreover, the clinical information for patients with schizophrenia with XPO7 variants, including detailed case reports, is important for further understanding the pathogenesis of XPO7 deficiency. We hypothesize the presence of a progressive molecular pathogenesis associated with long DUP and relapse, whereas large-scale analyses of patients with schizophrenia reveal evidence of cognitive impairment at the onset, but this impairment is stable throughout the course of the disease with appropriate treatment (McCutcheon et al, 2023; Velthorst et al, 2021). The treatment responsiveness as well as the cognitive and social course of patients with XPO7 variants are necessary for further interpretation of our findings. (Line 412-420)

[Reviewer's comment 5]

To identify molecules with abnormal nuclear import or export in the frontal cortex of mutant mice, the authors use IP assays followed by LC-MS/MS analysis to isolate XPO7-interacting proteins whose abundance is altered in nuclei isolated from the frontal cortex of mutant mice. Use of homozygous mutant mice is commendable as it leads to a well-controlled experiment. However, the authors analyze their IP data using two analytical methods and two seemingly arbitrary cutoffs to identify 43 and 412 proteins respectively. The rationale behind these two analytical approaches and why there is an order of magnitude difference in the number of identified protein targets is not clearly stated. It is also unclear why they have to present both analytical methods.

[Answer]

Thank you very much for your comments. To improve the rationale and logic of the extraction of the 45 molecules, or in other words, the rationale of using CutC as a cutoff marker for IP binding, we added the new results of the LC-MS/MS analysis of the nuclear extraction of XPO^{-/-} to **Figure 2B**. Among the proteins detected in the nuclear fraction of XPO^{-/-}, CutC was the most notably decreased (Figure 2B). The results were also reproduced using western blot analysis (Figure 2D). Based on these results and previous findings on the binding of Xpo7 and CutC, we set the IP binding cut-off value of CutC as a marker to identify candidate molecules that bind to CutC in immunoprecipitation experiments using Xpo7^{-/-} mice and an anti-Xpo7 antibody.

Initially, we searched for molecules that were most affected in the nuclear fraction of the frontal cortex of Xpo7^{-/-} mice by LC-MS/MS analysis. Among the proteins detected in the nuclear fraction of the frontal cortex of Xpo7^{-/-} mice, CutC was the most notably decreased (Fig. 2B). LC-MS/MS analysis also revealed a significant reduction (approximately half) in Xpo7^{+/-} mice compared with that in Xpo7^{+/+} littermates (Fig. 2C). Western blot analysis also confirmed the marked reduction of the CutC protein levels in the frontal cortex cell nuclei in Xpo7^{-/-} mice and Xpo7^{+/-} mice (Fig. 2D). These results indicate that CutC is transported into the nucleus by Xpo7 and is highly affected by Xpo7 haploinsufficiency.

(Line 147-155)

To identify molecules that bind to Xpo7, we calculated a ratio by dividing the amount of each molecule that was immunoprecipitated by the anti-Xpo7 antibody from the frontal cortex of Xpo7^{+/+} mouse by that of Xpo7^{-/-} mice. Because the ratio for CutC was 1.7, we considered molecules that interacted with Xpo7 as those exhibiting a ratio equal to or exceeding 1.7. Consequently, we discovered 412 molecules that bind to Xpo7 (Fig 2E). Of these, 45 showed significant ($p < 0.05$) changes in protein levels by LC-MS/MS analysis of the frontal cortex cell nuclei of Xpo7^{+/-} mice compared with the Xpo7^{+/+} mice (Fig. 2E). (Line 157-164)

[Reviewer's comment 6]

Combination of IP and LC-MS/MS analysis identified 45 molecules interacting with XPO7. Only a small fraction of them (only two, CutC and Gria3, if I understand it correctly) were followed up by WB assays. Confirming a larger fraction will provide more confidence in the list of identified targets.

[Answer]

We added additional western blot data for Rbfox3, Slitrk5 and IGSF21 (New Fig 3D, Fig3F, EV7B). The results further confirmed those from the LC-MS/MS analysis. Although it is difficult to validate all 45 molecules by western blot analysis because there are not always available antibodies, data-independent acquisition mass spectrometry (DIA-MS) (also known as SWATH-MS) conducted in this study is reliable in terms of quantification and reproducibility (Guzman et al, 2024; Kawashima et al, 2019; Shao et al, 2015). Furthermore, to reduce the number of foreign peaks and increase the peak selectivity of MS/MS, we narrowed the isolation window width in DIA-MS to 4 Th compared with the conventional 10–20 Th. This enables a more precise quantification of the identified proteins. In addition, high peak selectivity in MS/MS analysis is considered superior to western blot analysis for quantitative accuracy (Aebersold et al, 2013; Liebler & Zimmerman, 2013). In fact, we confirm that xpo7 was halved in *Xpo7^{+/-}* mice by DIA-MS.

We have described these issues in the Discussion section and Methods section as follows.

Although it is difficult to validate all 45 molecules by western blot analysis because there are not always available antibodies, data-independent acquisition mass spectrometry (DIA-MS) (also known as SWATH-MS) conducted in this study is reliable in terms of quantification and reproducibility (Guzman et al, 2024; Kawashima et al, 2019; Shao et al, 2015). Furthermore, we narrowed the isolation window width (see methods section), which enables more precise quantification. High peak selectivity in MS/MS analysis is recognized as superior to western blot analysis for quantitative accuracy (Aebersold et al, 2013; Liebler & Zimmerman, 2013). (Line 351-358)

DIA-MS with a narrow isolation width of 4 Th compared with the conventional 10–20 Th enables more precise quantification of the identified proteins. High peak selectivity in the MS/MS analysis is considered superior to the western blot analysis for quantitative accuracy (Aebersold et al., 2013; Liebler & Zimmerman, 2013). (Line 661-664)

[Reviewer's comment 7]

The age of the mice used in these analyses needs additional clarity. The authors state that mouse cerebral frontal cortex from XPO7^{+/+} and XPO7^{-/-} mice at the age of 1 month

were used for the IP assay. For the LC-MS/MS assay nuclei were isolated from the frontal cortex of 1-, 3-, and 6-months old mice. It is unclear how data from various age mice were used in individual analyses and the comparison between IP and LC-MS/MS data.

[Answer]

Because $Xpo7^{-/-}$ mice are lethal up to approximately one month of age, the immunoprecipitation analysis was conducted at the age of one month. For LC-MS/MS, three $Xpo7^{+/+}$ mice and three $Xpo7^{+/-}$ mice at one month of age were combined as one sample each to determine the relative change of $Xpo7^{+/-}$ mice to $Xpo7^{+/+}$ mice. Similarly, three $Xpo7^{+/+}$ mice and three $Xpo7^{+/-}$ mice at three and six months of age were also analyzed to determine the relative change of $Xpo7^{+/-}$ mice to $Xpo7^{+/+}$ mice. Consequently, the relative change of $Xpo7^{+/-}$ mice to $Xpo7^{+/+}$ mice at 1, 3, and 6 months were statistically analyzed to identify the molecules that change consistently in $Xpo7^{+/-}$ mice regardless of their age in months. Because we used one-month-old mice for IP analysis and three- and six-month-old mice for single-nucleus RNAseq analysis and behavioral experiments, we performed these cross-age analyses using LC-MS/MS. We describe the details in the Methods section and each of the figure legends as follows.

*LC-MS/MS analysis showing the relative expression of CutC in nuclei and whole cells from the frontal cortex of $Xpo7^{+/+}$ and $Xpo7^{+/-}$ mice. $**p < 0.01$ ($N = 3$ experiments, 3 mice per group/experiment; Tukey's HSD test). As described in Fig. 1C, three $Xpo7^{+/+}$ mice and three $Xpo7^{+/-}$ mice at 1 month of age were combined as one sample each to obtain the relative change of $Xpo7^{+/-}$ mice to $Xpo7^{+/+}$ mice. Similarly, three $Xpo7^{+/+}$ mice and three $Xpo7^{+/-}$ mice at three and six months of age were also analyzed to determine the relative change of $Xpo7^{+/-}$ mice to $Xpo7^{+/+}$ mice. Consequently, the relative change of $Xpo7^{+/-}$ mice to $Xpo7^{+/+}$ mice at 1, 3, and 6 months were statistically analyzed to identify the molecules that change consistently in $Xpo7^{+/-}$ mice regardless of their age in months.*

(Line 1030-1039)

As described in Fig. 1C and Fig 2C, three $Xpo7^{+/+}$ mice and three $Xpo7^{+/-}$ mice at 1 month of age were combined as one sample each to obtain the relative change of $Xpo7^{+/-}$ mice to $Xpo7^{+/+}$ mice. (Line 1054-1056)

As described in (A), the LC-MS/MS analysis data were derived from pairs of mice at 1, 3, and 6 months of age. (Line 1068-1069, 1075-1076)

[Reviewer's comment 8]

Identification of Gria3, an AMPA receptor subunit and one of the top 10 SCHEMA genes, as a target of XPO7 haploinsufficiency is very interesting and potentially one of the most important findings of this study as it could nicely demonstrate interaction between two top SCHEMA genes. The authors observed elevated levels of Gria3 within nuclei in the frontal cortex of mutant mice via LC-MS/MS analysis, while the extranuclear Gria3 levels were decreased at 6 months. The authors need to explain more clearly what the increase in nuclear levels of Gria3 (as opposed to decrease) imply. Also why only 6-month-old mice were analyzed? It is important to know if such changes appear earlier as well. Although I understand this may be outside the scope of this work, I think the value of the paper will increase immensely if the authors could probe the function of AMPA receptors in these mice using electrophysiological assays.

[Answer]

We are grateful for the reviewer's understanding of the importance of our study. We added new data for Gria3 levels at one and three months of age in Fig. 3B and Fig. EV7A. These data confirmed that the Gria3 levels in the extranucleus are decreased across all age groups. As the reviewer pointed out, we added that we hypothesized that the increase of Gria3 in the nuclear fraction resulting from nuclear export dysfunction and that the total Gria3 amount was similar between *Xpo7^{+/-}* mice to *Xpo7^{+/+}* mice indicates the reduction of the amount of Gria3 outside the nucleus, including the membrane fraction where Gria3 functions. This was confirmed by western blot analysis as described above. We described these as follows.

*We observed elevated levels of Gria3 within the nuclei of frontal cortical cells in *Xpo7^{+/-}* mice through LC-MS/MS analysis (Fig. 3A). Because the total Gria3 amount is compatible between *Xpo7^{+/-}* mice to *Xpo7^{+/+}* mice, we hypothesized that the extranuclear Gria3 levels are reduced, including the membrane fraction where Gria3 functions. We quantified the extranuclear Gria3 levels by western blotting analysis and confirmed a decrease in Gria3 outside the nucleus at 1, 3, and 6 months of age (Fig. 3B, Fig. EV7A). (Line 177-183)*

Regarding the AMPA receptor function, electrophysiological experiments are important for elucidating the pathophysiology. However, as the reviewer indicated, this is outside the scope of this work, as such an experiment should not be accomplished in a relatively short time. In future studies, we are planning to conduct electrophysiological experiments

in various brain regions, including the analysis of the compensatory mechanism by AMPA receptor subunits other than Gria3. Therefore, we would like to indicate the importance of electrophysiological experiments for future studies in the discussion as follows.

In addition, an electrophysiological study is essential in the future to further elucidate the effect of the reduction of the extranuclear Gria3 level in Xpo7 haploinsufficiency pathogenesis. (Line 334-336)

[Reviewer's comment 9]

There is no information on gross brain morphology and overall layer-specific density of neurons in adult mutant mice. Also there is no information on the impact of the XPO7 mutation on dendrites, and spines and axons of cortical neurons. Such data from morphological analysis will help interpret the single-nucleus RNA sequencing data. Along these lines the authors could use single-nucleus RNA sequencing data to determine and compare cell densities between wt and mutant mice.

Thank you very much for the comments. We added the analysis of adult brains at 3 months of age. There was no obvious change in brain size in hematoxylin and eosin (H&E) staining, and no apparent disturbance of the cortical layer structure by immunohistochemical analysis using layer markers between Xpo7^{+/+} and Xpo7^{+/-} mice (new Fig. EV2A–B). Consistent with these results, there were no obvious changes in the proportion of neurons in the cortical layers in the snRNA seq analysis (new Fig. EV8). We also performed Golgi staining to analyze the dendrites and spines. A significant decrease in the apical and basal dendrites of L2/3 neurons and the apical dendrites of L5/6 neurons in the frontal cortex was observed (Fig. EV3A–B). There were no significant differences in spine density per dendrite length (Fig. EV4A–B). We describe these changes in the Results section as follows.

There was no obvious change in brain size in hematoxylin and eosin (H&E) staining, and no apparent disturbance of the cortical layer structure by immunohistochemical analysis using layer markers between Xpo7^{+/+} and Xpo7^{+/-} mice (Fig. EV2A–B). We also performed Golgi staining to analyze the dendrites and spines. A significant decrease in the apical and basal dendrites of L2/3 neurons and the apical dendrites of L5/6 neurons in the frontal cortex was observed (Fig. EV3A–B). There were no significant differences in spine density per dendrite length (Fig. EV4A–B). (Line 110-116)

These changes in gene expression and nuclear translocation of molecules may be associated with the changes in the morphology of neurons. We observed a significant decrease in the apical and basal dendrites of L2/3 neurons and the apical dendrites of L5/6 neurons in the frontal cortex in $Xpo7^{+/-}$ mice. Although there were no significant differences in spine density per dendrite length by Golgi staining analysis, a decrease in dendrites suggests a total decrease of synapses in $Xpo7^{+/-}$ mice. Further analyses are necessary to determine the synaptic pathologies of $Xpo7^{+/-}$ mice. (Line 379-389)

There was no obvious difference in the ratio of neurons in the cortical layers between $Xpo7^{+/+}$ mice and $Xpo7^{+/-}$ mice (Fig. EV8) (Line 225-226)

Golgi staining and analysis

The mice were deeply anesthetized with ketamine (100 mg/kg, i.p.) and xylazine (10 mg/kg, i.p.) and then perfused with a formaldehyde/glutaraldehyde fixative (BioEnno, Irvine, CA). After perfusion, the brains were removed, post-fixed overnight in the same fixative, sectioned into 60 μm slices using a vibratome (Leica), and the floating sections were collected in PBS. On-slice Golgi staining was conducted using a SliceGolgi kit (BioEnno, Irvine, CA) based on the manufacturer's protocol, which involved a 7-day Golgi impregnation. After staining, the slices were mounted onto glass slides, dried, dehydrated in 100% ethanol, cleared with xylene, and coverslipped in Permount. Images were captured using an inverted light microscope (BZ-X710, Keyence) with a 10 \times objective lens for dendritic arborization analysis and a 100 \times objective lens for spine density and size analysis. Pyramidal neurons were selected from layers 2–3 or layers 5–6 of the frontal cortex, including the cingulate, infralimbic, prelimbic, and dorsal peduncular cortices.

The Golgi-stained images were analyzed using ImageJ software (<http://rsb.info.nih.gov/ij/>). Dendritic arborization was assessed using Sholl analysis with the Simple Neurite Tracer (SNT) plugin (ref 1). Neurites were traced along the z-axis for optimal accuracy and the Soma and dendrites were reconstructed using the ROI manager tool. Sholl analysis was performed with the shells set at a 10- μm radius and the intersections at each radius were counted. For each animal, 6–12 pyramidal neurons from either layers 2–3 or layers 5–6 were analyzed and the mean number of intersections was used as the representative value.

Spine density and size analysis included only pyramidal neurons with clearly identifiable spines and dendrites. Spine density was standardized by the length of the dendrites for comparative analysis. The spine head diameter was measured at the secondary and

higher-order dendrites of both the apical and basal dendrites, and defined as the minor axis of an ellipse-approximated spine head. Measurements were taken from 6–14 neurons in each dendrite per animal and the mean spine density and spine head diameter were considered representative values for each animal. (Line 558-585)

Based on the observation that there were no gross changes in the white matter by H&E staining, we presumed that there were no apparent abnormalities in the axon, but this is not a sufficient analysis. Because it is difficult to analyze axons in detail in vivo, we did not address this speculation in the manuscript; however, we described the importance of the analysis of axons and white matter in the Discussion as follows.

In this study, we focused on analyzing neurons; however, the glial cell pathology, including the pathology of white matter, is also important to schizophrenia pathology. Detailed analysis of glial cells and white matter together with axons will be the focus of future studies. (Line 386-389)

[Reviewer's comment 10]

Statements such as "we detected 284 gene expression changes that were linked to the behavioral phenotypes of XPO7" or "identified 15 genes in the frontal cortex related to disease progression" are inaccurate and require rephrasing. These are simply correlations as there is no experimental evidence linking these DEGs to either behavior or disease progression.

[Answer]

Thank you very much for your comment. As the reviewer pointed out, we clearly stated that these are simply correlations as described below.

In addition, through single-nucleus RNA sequencing of the frontal cortex and striatum of XPO7 haploinsufficiency mice differentiating between the onset and progressive stages, we identified 284 gene expression changes that correlated with these stages. (Line 31-34)

Furthermore, our approach identified 15 gene expression changes in the frontal cortex that correlated with the progressive stages. (Line 36-38)

[Reviewer's comment 11]

What is the rationale behind seeking overlaps between DEGs and XPO7 direct-effect

molecules? If anything, such overlap confounds interpretation of the direct effect analysis.

[Answer]

We speculated that a significant overlap between Xpo7 direct-effect molecules and DEGs suggests a compensatory response of DEGs because of the Xpo7 haploinsufficiency. Such a compensatory response by significantly overlapped molecules further strengthens the rationale of our results. However, as the reviewer pointed out, this overlap analysis may be redundant because such compensatory reactions can complicate the interpretation. Furthermore, these DEGs are only in part of the cell groups and are not significant in the bulk analysis. Therefore, in the revised manuscript, we deleted the description of the overlap between XPO7 direct-effect molecules and the DEGs.

Other issues:

[Reviewer's Comment 12]

It is not clear why the authors normalized all WT values to 1 in the relative expression data sets.

[Answer]

As described in the answer to comment 7, the WT values were set to 1 because the ratio of the average Xpo7^{+/-} value to the average WT value at one month of age and the ratios at three and six months of age were statistically analyzed. We described this in detail in the figure legend as follows.

three Xpo7^{+/+} mice and three Xpo7^{+/-} mice at one month of age were combined as one sample each to obtain the relative change of Xpo7^{+/-} mice to Xpo7^{+/+} mice. Similarly, three Xpo7^{+/+} mice and three Xpo7^{+/-} mice at three and six months of age were also analyzed to determine the relative change of Xpo7^{+/-} mice to Xpo7^{+/+} mice. Consequently, the relative change of Xpo7^{+/-} mice to Xpo7^{+/+} mice at 1, 3, and 6 months were statistically analyzed to identify the molecules that change consistently in Xpo7^{+/-} mice regardless of their age in months.

[Reviewer's comment 13]

The infomap of the PPI network analysis shown in Fig. EV2A is too small and it is impossible to read the single network targets.

[Answer]

We have uploaded a higher-resolution figure (new Fig EV6).

[Reviewer's comment 14]

It is not really clear what role, if any, CutC and potential copper-induced damage can have in the development and progression of schizophrenia as discussed in the result section. Is there some relevant evidence in the literature?

[Answer]

Although the reduction of CutC does not change the copper concentration, it decreases protection against damage from copper, which is similar to a condition of increased copper toxicity. A typical disease with copper toxicity is Wilson's disease, in which patients present with neurological symptoms with reported cases of psychosis (An et al, 2022; Grover et al, 2014). Excess copper causes oxidative stress (An et al., 2022). Because the pathogenetic effect of oxidative stress is associated with a variety of neurological disorders, this may also be related to schizophrenia (Cuenod et al, 2022). Furthermore, because copper concentrations are physiologically high in the locus coeruleus and substantia nigra, these regions may be affected by a reduction of CutC and reduced protection against damage from copper (Rihel, 2018; Szerdahelyi & Kása, 1986). Although further analysis is necessary, we describe this information in the Discussion section as follows.

Decreased protection against damage from copper may be similar to a condition of increased copper toxicity. A typical disease with copper toxicity is Wilson's disease, in which patients present with neurological symptoms with reported cases of psychosis (An et al, 2022; Grover et al, 2014). Excess copper causes oxidative stress (An et al., 2022). Because the pathogenetic effect of oxidative stress is associated with a variety of neurological disorders, this may also be related to schizophrenia (Cuenod et al, 2022). Furthermore, because copper concentrations are physiologically high in the locus coeruleus and substantia nigra, these regions may be affected by a reduction of CutC and reduced protection against damage from copper (Rihel, 2018; Szerdahelyi & Kása, 1986).

(Line 309-318)

[Reviewer's comment 15]

Why does the Ethics section refer to human subjects?

[Answer]

We have deleted the section pertaining to human subjects.

Referee #2:

[Reviewer's comment]

Toyoda et al. describe the generation of exportin 7, encoded by Xpo7, haploinsufficient and KO mice. XPO7 is a nuclear transport protein, identified as one of 10 SCHEMA (Schizophrenia Exome Sequencing Meta-Analysis) genes, carrying significant risk for schizophrenia. Two SCHEMA genes have been previously evaluated in mouse models, namely SETD1A and GRIN2A. This report will be the third SCHEMA gene evaluated in a mouse model. Here, Xpo7 haploinsufficient mice undergo behavioral and genomic analyses as a schizophrenia model. The authors find modest cognitive behavioral differences, which progress from adolescence to maturity. The genomics are the strength of the paper. The authors use intersectional approaches in single cell analyses to identify transported proteins that are differentially found in the nucleus, in the Xpo7 haploinsufficient mice. They find differentially regulated genes, which include other SCHEMA genes. Looking at the intersection of Xpo7 and progression-associated genes is particularly powerful, revealing an overlap with 4 of 10 SCHEMA genes. The authors perform cell-type and regional (cortex and striatum) analyses to localize gene expression changes. While no one mutant mouse can model schizophrenia fully, this study instantiates several associations that support the pathogenic schizophrenia risk of XPO7 deficiency.

[Reviewer's Comment 1]

Description of the generation of the XPO7 mutant mice is cursory. A gene map for their generation would be expected as Fig 1A. Description of the mice indicates that exon 7 is deleted. Is a truncated protein expressed from exons 1-6? Could this confer a toxic gain of function?

[Answer]

Thank you very much for your comment. We added a new figure for the gene map to clarify that exon 7 has been deleted (new Fig. 1A). Xpo7 contains 1088 amino acids (aa) and exons 1–6 correspond to aa 1–199, aa 200–255 for exon 7, and aa 256–1088 for exons 8–28. Because the antigen for the anti-Xpo7 antibody used in the western blot in Fig. 1B recognizes 520–594 aa, exons 7–28 are not expressed as proteins. In LC-MS/MS, six types of peptide fragments were detected from the sequence including 1–199 aa (exons 1–6) from Xpo7^{+/+} and Xpo7^{+/-} mice, but not from Xpo7^{-/-} mice (Fig. EV1B). Collectively, these data suggest that the truncated proteins of exons 1–6 are not expressed in the Xpo7^{+/-} and Xpo7^{-/-} mice analyzed in this study. The removal of a single exon

from the genome often results in total knockout, probably because of genomic disturbance. This point is important, so we have included it in the Results as follows.

Xpo7 is a 1088 amino acid (aa) molecule, and exons 1–6 correspond to 1–199 aa, exon 7 to 200–255 aa, and exons 8–28 to 256–1088 aa. Because the antigen for the anti-Xpo7 antibody used in the western blot in Fig. 1B recognizes 520–594 aa, exons 7–28 are not expressed as proteins. In LC-MS/MS, six types of peptide fragments were detected from the sequence including 1–199 aa (exons 1–6) from Xpo7^{+/+} and Xpo7^{+/-} mice, but not from Xpo7^{-/-} mice (Fig. EV1B). Collectively, these data suggest that the truncated proteins of exons 1–6 are not expressed in the Xpo7^{+/-} and Xpo7^{-/-} mice analyzed in this study. (Line 101-108)

[Reviewer's comment 2]

It should be discussed that Xpo7 is a nuclear exportin, and that the majority of its activity is nuclear export, not import.

[Answer]

Thank you very much for your comment. As the reviewer pointed out, Xpo7 is an exportin and its primary function is considered to be nuclear export. In fact, among the 45 molecules that we found as Xpo7 direct-effect molecules, 33 were upregulated in the of the frontal cortex cell nuclei of Xpo7^{+/-} mice, indicating that these molecules are affected by the nuclear export deficiency of Xpo7 haploinsufficiency. On the other hand, CutC, which we identified as the molecule most affected by Xpo7 deficiency in Xpo7^{+/-} mice, was reduced in the nucleus, indicating that Xpo7 is involved in the nuclear import of CutC. Previous studies have also reported that Xpo7 functions both as an exportin and an importin (Aksu et al., 2018). Because this point is important in terms of Xpo7 function, we have described this in the discussion section as follows.

As its name suggests, the basic function of Xpo7 is nuclear export. In fact, among the 45 molecules that we found as Xpo7 direct-effect molecules, 33 were upregulated in the of the frontal cortex cell nuclei of Xpo7^{+/-} mice, indicating that these molecules are affected by the nuclear export deficiency of Xpo7 haploinsufficiency (Fig. 2F). On the other hand, CutC, which we identified as the molecule most affected by Xpo7 deficiency in Xpo7^{+/-} mice, was reduced in the nucleus, indicating that Xpo7 is involved in the nuclear import of CutC (Fig. 2B, 2C, and 2D). Previous studies have also reported that Xpo7 functions

both as an exportin and an importin (Aksu et al., 2018). Our results further show that Xpo7 has both functions of nuclear export and import. (Line 288-296)

[Reviewer's comment 3]

Methods are lacking for assessing gene expression (allelic abundance), shown in Fig 1B, as well as genotyping.

[Answer]

The primer sequences and methods for genotyping PCR are described as follows. The details of the LC-MS/MS analysis in **new Fig 1C** (previously Fig 1B) were added to the figure legend and methods as follows.

Genotyping Polymerase Chain Reaction (PCR) of Xpo7 knockout mice.

The primer, template, and enzyme used were as follows. Forward Primer: 5' -

GCCACACCCCATATTGCTTATG - 3', Reverse Primer: 5' -

GGAACCACAGGTTAAGTGACAC - 3'. Template: DNA extracted from the ear by the alkaline lysis method. Enzyme: TaKaRa Ex Premier™ DNA Polymerase (Takara). The annealing temperature was 57°C and the others steps were performed per the manufacturer's protocol of the enzyme. (Line 521-527)

*The relative expression of Xpo7 by proteome analysis of the frontal cortex in Xpo7^{+/+} and Xpo7^{+/-} mice. **p < 0.01 (N = 3 experiments, 3 mice per group (1, 3, 6 months of age)/experiment; Tukey's HSD test). Specifically, three Xpo7^{+/+} mice and three Xpo7^{+/-} mice at 1 month of age were combined as one sample each to obtain the relative change of Xpo7^{+/-} mice to Xpo7^{+/+} mice. Similarly, three Xpo7^{+/+} mice and three Xpo7^{+/-} mice at 3 and 6 months of age were also analyzed to determine the relative change of Xpo7^{+/-} mice to Xpo7^{+/+} mice. Consequently, the relative change of Xpo7^{+/-} mice to Xpo7^{+/+} mice at 1, 3, and 6 months were statistically analyzed to identify the molecules that change consistently in Xpo7^{+/-} mice regardless of their age in months. (Line 1002-1011)*

Sample preparation for proteome analysis by liquid chromatograph - mass spectrometry (LC-MS/MS)

Isolated nuclei from the frontal cortex (1, 3, and 6 months of age from Xpo7^{+/+} and

Xpo7^{+/-} mice, 1 month of age from *Xpo7^{-/-}* mice; 3 mice per group, prepared as above) and total frontal cortex samples (1, 3, and 6 months of age from *Xpo7^{+/+}* and *Xpo7^{+/-}* mice, 1 month from *Xpo7^{-/-}* mice; 3 mice per group) were further processed for analysis by LC-MS/MS as described in a previous study (Kawashima et al, 2022). In brief, cells were dissolved in 100 mM Tris-HCl (pH 8.0) containing 4% SDS and 20 mM NaCl using BIORUPTOR BR-II (SONIC BIO Co., Kanagawa, Japan). The Pierce™ BCA Protein Assay Kit (Thermo Fisher Scientific, WA, USA) at 500 ng/μL was quantified into 20 micrograms of extracted proteins. The protein extracts were reduced with 20 mM tris(2-carboxyethyl) phosphine for 10 minutes at 80 °C, followed by alkylation with 30 mM iodoacetamide for 30 minutes at room temperature in the dark. Protein purification and digestion were performed using the SP3 method (Hughes et al, 2019; Kawashima et al., 2022). Tryptic digestion was performed using 500 ng/μL Trypsin/Lys-C Mix (Promega, Madison, WI, USA) overnight at 37 °C. The digests were purified using GL-Tip SDB (GL Sciences, Tokyo, Japan) according to the manufacturer's protocol. The peptides were dissolved again in 2% ACN containing 0.1% TFA and quantified using the BCA assay at 150 ng/μL. (Line 608-626)

Liquid chromatography - mass spectrometry (LC-MS/MS)

For LC-MS/MS analysis, digested peptides from isolated nuclei from the frontal cortex and total frontal cortex samples were loaded directly onto a 75 μm × 12 cm nanoLC nanocapillary column (Nikkyo Technos Co., Ltd., Tokyo, Japan) at 40 °C and then separated with a 30-minute gradient (mobile phase A: 0.1% formic acid (FA) in water, mobile phase B: 0.1% FA in 80% acetonitrile (ACN)). The gradient was 0 min at 8% B, increasing to 70% B at 30 min, with a 200 nL/min flow rate using an UltiMate 3000 RSLCnano LC system (Thermo Fisher Scientific). The eluted peptides were detected using a quadrupole Orbitrap Exploris 480 hybrid mass spectrometer (Thermo Fisher Scientific) with a standard window data-independent acquisition (DIA). The MS1 scan range was set to a full scan with m/z 490–745 at a mass resolution of 15,000, an Auto Gain Control (AGC) target of 3×10^6 , and a maximum injection time of 23 milliseconds. The MS2 scans were collected for ions greater than m/z 200 at a resolution of 30,000, with an AGC target of 3×10^6 , a maximum “auto” injection time, and a fixed normalized collision energy of 28%. The isolation width for MS2 was set to 4 Th, and for the 500–740 m/z window pattern, an optimized window arrangement was used in Scaffold DIA (Proteome Software, Inc.,

Portland, OR, USA). DIA-MS with a narrow isolation width of 4 Th compared with the conventional 10–20 Th enables more precise quantification of the identified proteins. High peak selectivity in the MS/MS analysis is considered superior to the western blot analysis for quantitative accuracy (Aebersold et al., 2013; Liebler & Zimmerman, 2013). (Line 645-664)

[Reviewer's comment 4]

There is no description of the dissections, what part of cortex was taken, no details on how cortical layers were separated, nor criteria used for isolating the striatum.

[Answer]

We conducted single-nucleus RNA sequencing analysis at three and six months in the section containing both brain regions (Bregma 1.30–3.30, excluding the olfactory area). We did not isolate the striatum from the cortex by anatomical manual dissection. Because dSPN and iSPN in the striatum are inhibitory neurons that express the dopamine D1 and D2 receptors, and such neurons do not exist in the cortex, it is possible to dissect these neurons in silico from single-nucleus RNA sequencing data. Similarly, because the vast majority of neurons are inhibitory neurons in the striatum, subcluster analysis of cortical layer neurons is possible in silico from the data of single-nucleus RNA sequencing analysis. We added this information to the Methods section as follows. Also, we included the information that the striatum analyzed in our sections was cranial striatum based on the brain regions (Bregma 1.30–3.30, excluding the olfactory area).

*To determine the specific cell populations that are affected in the frontal cortex and striatum of $Xpo7^{+/-}$ mice, we conducted single-nucleus RNA sequencing analysis at 3 and 6 months in the sections containing both brain regions (Bregma 1.30–3.30, excluding the olfactory area) (Fig. 4A, 4B and Table EV1). The neurons in these sections can be dissected into subpopulations in silico using single-nucleus RNA sequencing data. For example, the vast majority of excitatory neurons from these sections originate in the frontal cortex and these neurons can be further identified by cortical layer markers (Fig. 4D and 4E). In addition, inhibitory neurons in these sections originate in the frontal cortex and striatum and can be further classified by the dopamine D1 receptor (*Drd1*) and dopamine D2 receptors (*Drd2*) for medium spiny neurons (dSPN and iSPN) in the striatum as well as other markers, such as parvalbumin (Fig. 6A and 6B). (Line 201-218)*

The cell nuclei were collected from the sections containing both the frontal cortex and

striatum (Bregma 1.30 – 3.30, excluding the olfactory area) of 12 mice [4 groups ($Xpo7^{+/+}$ and $Xpo7^{+/-}$ mice at 3 and 6 months of age); three mice/group]. (Line 714-716)

[Reviewer's comment 5]

Nomenclature wise, in mice XPO7 should be written Xpo7 (human genes are all caps, mouse genes only first letter is capitalized).

[Answer]

Thank you for the comment. We have corrected the descriptions.

[Reviewer's comment 6]

Page 10, sentence, "The 45 molecules are presented in Fig. 3F. Gene Ontology (GO) analysis of these 45 molecules revealed that they contain many receptors and transporters involved in synapses (Fig. 3G)." Figure citations should be to Fig. 2, not 3.

[Answer]

Thank you for the comment. We have corrected it.

[Reviewer's comment 7]

Page 11, sentence, "We observed elevated levels of Gria3 within the frontal cortex of the nucleus of XPO7+/? mice through LC-MS/MS analysis (Fig. 3A)." Would be better, ...within the nuclei of frontal cortical cells in Xpo7+./- mice through...

We have corrected the description.

[Reviewer's comment 8]

Page 11, sentence, "CutC. Additionally, Fyn (Odds ratio 10.7, $p = 0.000596$) and Slitrk5 (Odds ratio 6.69, $p = 0.000917$) were molecules linked to an increased susceptibility to schizophrenia (Singh et al., 2022)." It should be stated that these genes were upregulated.

[Answer]

We have added the description as follows.

Slitrk5 and Fyn in the nucleus are upregulated in $Xpo7^{+/-}$ mice. (Line 187-188)

[Reviewer's comment 9]

Page 12, sentence, "Additionally, striatal pathology has been frequently observed alongside dopamine and cortical pathology in schizophrenia." It is not clear what alongside means, nor is this statement referenced.

[Answer]

Thank you for the comment. We have corrected the description as follows.

The frontal cortex and the striatum are the brain areas that are significantly associated with schizophrenia. Cognitive function and social interactions are closely linked to the frontal cortex functionality. The dopamine pathology of schizophrenia is well known and the association between the striatum, which strongly expresses dopamine receptors, and schizophrenia has been repeatedly reported (Chand et al., 2020; McCutcheon et al., 2019; Simpson et al., 2010). (Line 201-207)

[Reviewer's comment 10]

Page 13, sentence, "This is crucial as XPO7^{+/+} mice exhibit normal cognitive function and sociability at 6 months of age." This seems obvious, as XPO7^{+/+} mice are wild type mice.

[Answer]

We have deleted the unnecessary sentences and simplified the description as follows.

We then used the following approach to identify the DEGs related to the pathology of Xpo7^{+/-} mice. It is important to note that among the DEGs in these excitatory neurons, we need to determine that the DEGs in Xpo7^{+/-} mice implicated at the onset of pathology at 3 months not only differ from age-matched controls, but also the controls at 6 months. Similarly, DEGs in Xpo7^{+/-} mice at 6 months of age should differ not only from age-matched controls, but also from controls at 3 months of age. Genes that meet these criteria are more closely related to the pathogenesis of Xpo7 haploinsufficiency. (Line 232-239)

[Reviewer's comment 11]

Page 21, sentence, "The best approach for treating XPO7 haploinsufficiency would be to restore the expression of XPO7 itself." Yes, if the restoration were done in early embryogenesis; there is no assurance that restoring gene function in maturity will reverse developmental pathogenic effects (i.e. shutting the barn door after the horse has bolted).

[Answer]

Thank you very much for your comments. This is an important point when considering the treatment of patients with XPO7 dysfunction. As the reviewer points out, to assess whether restoring Xpo7 expression is therapeutic after birth or onset, it is necessary to understand exactly when and to what extent Xpo7 pathology occurs by controlling the developmental stage or age-specific Xpo7 expression. Thus, we have revised the description as follows.

One approach for treating XPO7 haploinsufficiency is to restore the expression of XPO7 itself; however, further studies are needed to verify whether this approach can be used as a treatment. For example, determining whether behavioral abnormalities are present when Xpo7 expression is reduced only after birth will make it clear to what extent the Xpo7 haploinsufficiency results in pathogenesis after birth. Examining whether recovery is possible by restoring Xpo7 expression after birth or after the onset of the disease (after the age of three months) in Xpo7 haploinsufficiency mice will make it clear whether restoring the expression of Xpo7 is a viable treatment. In addition, the latter experiments reveal to what extent Xpo7 haploinsufficiency in embryogenesis can form behavioral and molecular pathologies. (Line 422-432)

[Reviewer's comment 12]

Westerns are presented cutup, with the dynamic range differentially adjusted, based on background shade. They should be shown whole.

[Answer]

We uploaded the whole western blotting data as source data.

[Reviewer's comment 13]

Fig 3A - should specify from 3 months old mice.

[Answer]

Thank you for the comment. As the reviewer points out, the LC-MS/MS data are cross-age analyzed. We have added the description in the figure legend as follows. To specify the analysis of Gria 3 from three-month-old mice, we added new western blot data from three-month-old mice (new Fig 3B).

Three Xpo7^{+/+} mice and three Xpo7^{+/-} mice at 1 month of age were combined as one

sample each to obtain the relative change of Xpo7^{+/-} mice to Xpo7^{+/+} mice. Similarly, three Xpo7^{+/+} mice and three Xpo7^{+/-} mice at 3 and 6 months of age were also analyzed to determine the relative change of Xpo7^{+/-} mice to Xpo7^{+/+} mice. Consequently, the relative change of Xpo7^{+/-} mice to Xpo7^{+/+} mice at 1, 3, and 6 months were statistically analyzed to identify the molecules that change consistently in Xpo7^{+/-} mice regardless of their age in months. (Line 1055-1061)

[Reviewer's comment 14]

Fig EV3 uses the same color scale for up and down regulation, and does not include a fold scale look-up table.

[Answer]

New Fig. EV10 (previously Fig. EV3) shows how much each group of DEGs is related to the GO terms listed on the right; thus, it is not a heat map showing up- or down-regulation. We added a figure with a color scale showing the significance of the relationship.

Dear Prof. Shiwaku,

Thank you for the submission of your revised manuscript. We have now received the enclosed reports from the referees and I am happy to say that both support its publication now. Referee 2 still has a few more minor suggestions that I would like you to incorporate before we can proceed with the official acceptance of your manuscript. Please co-submit a point-by-point response with your final ms.

A few editorial requests will also need to be addressed:

- Please remove the author credits from the ms file. All credits need to be entered during online ms submission.
- Please remove "data not shown" on page 25, as per journal policy.
- In the author checklist under "Data Availability", you do not mention that you deposited your single nucleus RNA seq data in a public domain. Actually, the DAS needs to be updated with a link from a formal repository; the link provided in the ms file is not accessible and depositing data in a dropbox is not ideal.
- The funding info in the ms and in our online submission system are not congruent: JPMJFR231U seems to be missing in the ms. Can you please double-check whether the info is matching on both ends and correct.
- Please upload all main and all EV figures and EV tables as individual, high resolution files. The legends of the EV figures need to be pasted to after the main figure legends in the ms file. The legend for the EV table needs to be part of the table file.
- Our Methods section now needs to include a Reagents and Tools Table (listing key reagents, experimental models, software and relevant equipment and including their sources and relevant identifiers). A downloadable templates (.docx) for the Reagents and Tools Table can be found in our author guidelines: <
<https://www.embopress.org/page/journal/14693178/authorguide#manuscriptpreparation>>.
- The source data need to be uploaded as 1 (if necessary zipped) folder per figure. Each folder can have several files for the figure panels.
- The manuscript sections should be in the following order: Title page - Abstract & Keywords - Introduction - Results - Discussion - Methods - Data Availability - Acknowledgments - Disclosure Statement & Competing Interests - References - Figure Legends - (Main Tables with legends if applicable) - Expanded View Figure Legends.

Please address the following *Figure Legends - Comments*

- Please note that the legend for figures EV 6c-f is mislabeled as EV 6b-e. This needs to be rectified.
- Please note that the exact p values are not provided in the legends of figures 1c-f; 2c; 3a-b, d, f-i; 4h; 5e; EV 7a-b, d; EV 10. Exact p-values need to be provided as reasonable.
- Please indicate the statistical test used for data analysis in the legends of figures 4c, f-g; EV 9.
- Please note that in figures 1f; 3c, e; EV 7a, c, e-f; there is a mismatch between the annotated p values in the figure legend and the annotated p values in the figure file that should be corrected.
- Please note that information related to n is missing in the legends of figures 4h; 5e; 6d-e; EV 10.
- Please note that the error bars are not defined in the legends of figures 1f-g; 3b-i; EV 3b; EV 4a-b; EV 5a-c; EV 7a-f.

I would like to suggest a few minor changes to the abstract that needs to be written in present tense. Please let me know whether you agree with the following:

Recent genetic studies by the Schizophrenia Exome Sequencing Meta-Analysis (SCHEMA) consortium have identified that protein-truncating variants of exportin 7 (XPO7) can increase the risk of schizophrenia (odds ratio, 28.1). Here we show that mice with Xpo7 haploinsufficiency (Xpo7^{+/-} mice) present with cognitive and social behavioral impairments. Through proteome analysis using immunoprecipitation and frontal cortex nuclear isolation of Xpo7^{+/-} mice, we identify 45 molecules interacting with Xpo7, including CutC, Rbfox3, and Gria3. Through single-nucleus RNA sequencing of the frontal cortex and striatum of Xpo7^{+/-} mice differentiating between the onset and progressive stages, we also identify 284 gene expression changes that correlate with these stages. These genes encompass high-odds risk genes of schizophrenia identified by SCHEMA, including Gria3, Grin2A, Herc1, and Trio. Furthermore, our approach reveals 15 gene expression changes in the frontal cortex that correlate with the progressive stages. Our findings indicate the importance of investigating whether the interactions among the high-risk genes identified by SCHEMA contribute to a common schizophrenia pathology and underscore the significance of stage-dependent analysis.

EMBO press papers are accompanied online by A) a short (1-2 sentences) summary of the findings and their significance, B) 2-3 bullet points highlighting key results and C) a synopsis image that is exactly 550 pixels wide and 200-600 pixels high (the height is variable). The synopsis image should provide a sketch of the major findings, like a graphical abstract. Please note that text needs to be readable at the final size. Please send us this information along with the final manuscript.

Referee #1:

I find the authors' responses to my comments satisfactory and believe they have adequately addressed the concerns raised.

Referee #2:

The revised ms. is significantly improved, with the addition of new data, better argued, and now suitable for publication, once a few minor points are addressed.

Minor points

1 Fig 4G: The linked DEG's below the bar graph are not explained here or subsequently in the legends.

2 It is not clear what is being argued in lines 355 - 362 paragraph of the Discussion

3 Line 471: Statement is added that behavioral test were performed using C57BL/6J mice (also all mice were male). Was the background of all mice studied C57BL/6J? Assuming so, how many times were mice backcrossed to achieve this?

4 537-539: In the Westerns, were protease inhibitors included in the lysis procedure?

5 In Fig 4G, the linked DEG's below the bar graph are not explained, nor subsequently in the legends.

6 There are minor grammatical problems in Lines 291-294, and line 338.

Referee #2:**[Reviewer's comment 1]**

Fig 4G: The linked DEG's below the bar graph are not explained here or subsequently in the legends.

[Answer]

The explanation for the linked DEGs has been added in the figure legend as follows:

The linked DEGs below the bar graph indicate that the DEGs corresponding to each bar belong to all the indicated rows. For example, 45 DEGs belong to both “DEGs 3M Xpo7^{+/-} vs 3M Xpo7^{+/+} up” and “DEGs 3M Xpo7^{+/-} vs 3M Xpo7^{+/+} up” (Line 1096-1099)

[Reviewer's comment 2]

It is not clear what is being argued in lines 355 - 362 paragraph of the Discussion

[Answer]

The corresponding paragraph was largely meant to serve as an explanation for the reviewer and was technical in nature. However, the corresponding and relevant information has also been provided in the Methods section, and hence, the paragraph has been deleted from the discussion section.

[Reviewer's comment 3]

Line 471: Statement is added that behavioral test were performed using C57BL/6J mice (also all mice were male). Was the background of all mice studied C57BL/6J? Assuming so, how many times were mice backcrossed to achieve this?

[Answer]

The description of the mice has been appended to the Methods section as follows.

Male mice with the C57BL/6J background were used for conducting all behavioral tests and biochemical analyses. The generated Xpo7^{+/-} mice were backcrossed more than five times with C57BL6/J mice prior to the analysis. (Line 455-457)

The generated Xpo7^{+/-} mice were backcrossed more than five times with C57BL6/J mice prior to the analysis. (Line 510-511)

[Reviewer's comment 4]

537-539: In the Westerns, were protease inhibitors included in the lysis procedure?

[Answer]

We apologize that the information regarding the protease inhibitor in the lysis buffer was not included earlier. The relevant information regarding the protease inhibitor Cocktail (1:200, 539134-1SETCN, Merck) has now been included in the Methods section.

[Reviewer's comment 5]

In Fig 4G, the linked DEG's below the bar graph are not explained, nor subsequently in the legends.

[Answer]

Please refer to the answer to Comment 2.

[Reviewer's comment 6]

There are minor grammatical problems in Lines 291-294, and line 338.

[Answer]

We have corrected the description as follows.

However, CutC, identified as the molecule that was most affected by Xpo7 deficiency in Xpo7^{+/-} mice, was reduced in the nucleus, thereby indicating that Xpo7 plays a role in the nuclear import of CutC. (Line 289-292)

The reduction of Rbfox3 in the nucleus is also a potential pathogenic factor. (Line 336)

Prof. Hiroki Shiwaku
Institute of Science Tokyo
Department of Psychiatry and Behavioral Sciences
1-5-45 Yushima
Bunkyo-ku
Tokyo, Tokyo 113-8510
Japan

Dear Prof. Shiwaku,

I am very pleased to accept your manuscript for publication in the next available issue of EMBO reports. Thank you for your contribution to our journal.

Yours sincerely,
